# Uncovering the dynamics of precise repair at CRISPR/Cas9-induced double-strand breaks

Daniela Ben-Tov[1,2], Fabrizio Mafessoni[1,2], Amit Cucuy [1], Arik Honig[1], Cathy Melamed-Bessudo [1]✉ & Avraham A. Levy [1]✉

CRISPR/Cas9 is widely used for precise mutagenesis through targeted DNA double-strand breaks (DSBs) induction followed by error-prone repair. A better understanding of this process requires measuring the rates of cutting, error-prone, and precise repair, which have remained elusive so far. Here, we present a molecular and computational toolkit for multiplexed quantification of DSB intermediates and repair products by single-molecule sequencing. Using this approach, we characterize the dynamics of DSB induction, processing and repair at endogenous loci along a 72 h time-course in tomato protoplasts. Combining this data with kinetic modeling reveals that indel accumulation is determined by the combined effect of the rates of DSB induction processing of broken ends, and precise versus error repair. In this study, 64–88% of the molecules were cleaved in the three targets analyzed, while indels ranged between 15–41%. Precise repair accounts for most of the gap between cleavage and error repair, representing up to 70% of all repair events. Altogether, this system exposes flux in the DSB repair process, decoupling induction and repair dynamics, and suggesting an essential role of high-fidelity repair in limiting the efficiency of CRISPR-mediated mutagenesis.

DNA DSBs are one of the most cytotoxic forms of DNA damage, both endangering genome stability and driving genome evolution, through error-prone repair. In recent years, the ability to target DSBs to specific endogenous sequences, using custom-designed nucleases, in particular Clustered Regulatory Interspaced Short Palindromic Repeat associated protein Cas9 (CRISPR-Cas9), has triggered a revolution in genome engineering[1–5], widely impacting the field of life science from biomedical technology and research[6–10] to agriculture[11–15].

Optimization of this technology for precise genome engineering requires understanding DSB induction and the dynamics of repair by the endogenous machinery. Once a DSB is made, it is repaired by a complex network of highly conserved pathways, which can be characterized by the scars they leave behind, such as insertions and/or deletions (indels) or conversion tracts resulting from templated repair[16]. These pathways, generally subdivided into non-homologous end-joining (NHEJ) and homologous recombination (HR), have been extensively studied in organisms from yeast to plants and mammals[16–23]. While canonical NHEJ (cNHEJ) is the primary pathway in somatic plant and mammalian cells, other NHEJ pathways can act at a DSB site, involving micro-homologies and/or DNA synthesis, when direct re-ligation through the canonical pathway has either failed or is unavailable[24–26]. These pathways have been referred to as alternative-EJ (alt-EJ), microhomology-mediated end-joining (MMEJ), or Polymerase Theta Mediated End-Joining (TMEJ). In addition, our lab and others have shown that HR can act in somatic cells to repair DSBs through inter-homologue recombination[27–31], or can lead to chromosomal rearrangement[12,32].

Since DNA is constantly exposed to damage[33], maintaining genome integrity and limiting the rapid accumulation of somatic mutations would require high-fidelity repair[34]. Accumulating evidence supports the suggestion that NHEJ may be highly accurate in mammalian systems[34–38]. For example, DSBs induced by *I-SceI* were shown to be repaired precisely up to ~75% of the time in mouse cells[38]. In contrast, work characterizing re-ligation of linearized plasmids in both

---

[1]Department of Plant and Environmental Sciences, Weizmann Institute of Science, Rehovot 76100, Israel. [2]These authors contributed equally: Daniela Ben-Tov, Fabrizio Mafessoni. ✉e-mail: cathy.bessudo@weizmann.ac.il; avi.levy@weizmann.ac.il

tobacco explants and in protoplasts suggests that NHEJ is highly error-prone, often displaying evidence of 'filler DNA'[17]. Direct comparison of repair-fidelity in plants and mammals supports these results, measuring 50-55% precise repair in HeLa cells, compared to only 15-30% in tobacco cells[39]. However, these studies were largely conducted in the context of exogenous or transgenic DNA.

Studies of DSBs induced by CRISPR/Cas9 in endogenous chromatin have primarily focused on mutagenic repair, revealing many aspects affecting DSB repair outcome and repair pathway choice[40,41]. In addition, kinetic studies have provided insight into the dynamics of induction and error-prone repair[41-44]. However, due to the inability to directly observe scar-less re-ligation, measuring the degree of precise repair, particularly by the end-joining pathways, remains challenging. In addition, few studies follow the process of DSB induction and the intermediates between break and repair, namely the processing of the DSBs that leads to specific types of outcomes. Despite efforts, much of the process of CRISPR/Cas9-induced DSB repair remains unknown, particularly in somatic plant cells, including the efficiency and timing of induction, characteristics of DSB intermediates that lead to the different outcomes, and the fidelity of repair.

Resolving precise repair at CRISPR/Cas9 induced DSBs, in endogenous chromatin, requires quantification of both intermediates and products along a time-course, facilitating estimation of the rates of cutting and repair, both precise and error-prone, through mathematical modeling. Using measurements of repair-error, quantified through amplicon sequencing, and DSB formation through ligation-mediated PCR, one study estimated cutting and repair rates in four CRISPR-Cas9 targets in K562 cell lines[44]. Their results suggest that repair is often slow and highly error-prone, with an estimated half-life of up to ~10 h. However, this approach is limited by resolution achievable in quantifying both DSBs and repair products, leading to estimates of precise repair accompanied by a large degree of uncertainty[44] Reducing the uncertainty in the rate constant estimations requires improving the precision and resolution of measuring both DSBs and products of repair error.

Here, we present UMI-DSBseq, a molecular and computational toolkit for direct quantification of DSB intermediates alongside repair-products through multiplexed sequencing. The use of unique molecular identifiers (UMIs) enables the measurement of DNA cleavage and repair events at the single-molecule resolution. Using this approach, we follow the dynamics of DSB induction, end-processing, and repair at three CRISPR/Cas9 targets in tomato protoplasts, measuring the abundance of DSB intermediates and repair products simultaneously along a 72 h time-course. In addition to revealing characteristics of DSB intermediates, including evidence for dual cleavage, combining this data with kinetic modeling reveals rapid DSB induction and precise restoration of the original DNA sequence as prominent features of all repair events. Finally, we follow the processing of unrepaired DSBs, revealing the contribution of processed intermediates to error-prone repair. Altogether, these results suggest that CRISPR-mediated indels frequency following DSB induction in plants is determined as much by the fidelity of the endogenous repair process as by the ability to efficiently induce DSBs at the target site.

## Results

### Single-molecule quantification and characterization of DSB intermediates and error-prone repair products using UMI-DSBseq

Evaluating the process of DSB repair, from intact to final repair-outcome, requires controlled, relatively synchronized DSB induction, followed by quantitative characterization of both DSB-repair intermediates and repair-products. To this end, preassembled ribonucleoproteins (RNPs), consisting of SpCas9 protein and synthetic sgRNA, are delivered directly into plant protoplasts purified from tomato seedlings using PEG-mediated transformation (Fig. 1A). This method, which

does not rely on transcription, translation by the endogenous machinery, or assembly of Cas9 with the sgRNA in vivo, bypasses the lag-time between introduction and activity, allowing for fast induction of DSBs in a controlled manner. Following RNP delivery, we can follow dynamics of DSB induction and repair process along a time-course.

To address the challenge of directly quantifying the proportions of molecules, broken and repaired, we designed UMI-DSBseq, a ligation-mediated PCR-based assay combining a target-specific primer with a DSB flanking restriction enzyme site to capture both DSBs and intact molecules (Fig. 1B). The UMI-DSBseq assay facilitates high-resolution characterization of the state of all molecules by simultaneous ligation of adaptors containing unique molecular identifiers (UMIs) directly to unrepaired DSBs, as well as to intact molecules (both WT and containing indels) at the flanking restriction enzyme site, cleaved in vitro (Fig. 1B, C). To ensure the capturing of all molecules in the pool, including any potentially resected molecules, ends are repaired by fill-in of 3' prior to adaptor ligation. The UMI-DSBseq protocol facilitates direct preparation of Illumina sequencing-ready libraries, enabling multiplexed sequencing of unrepaired DSBs in conjunction with repair products, all in a single tube (Fig. 1C). Following sequencing, molecules can be categorized as either unrepaired DSBs, WT intact molecules, or indel products of error-prone repair (www.github.com/daniebt/UMI-DSBseq). A comparison of UMI-DSBseq and PCR-direct (primers shown in Supplementary Data 1) shows a similar ability in capturing the proportion of indels generated after exposure of molecules to CRISPR/Cas9 (Fig. S1).

### Capturing the kinetics of DSB induction and repair at endogenous loci

Three targets were selected for study in tomato, *Solanum Lycopersicum* cv. M82., at exons of three genes: *Carotenoid Isomerase*, *CRTISO*[45], *Phytoene Synthase 1*, *Psy1*[27] and *Phytochrome B2*, *PhyB2*[46] (for oligos see Supplementary Data 1). Time-courses consisting of 7 time-points, were collected for each target with 2 replicates independently transformed for each time-point (Fig. 2, for data see Source Data). Each sample collected for analysis contains a mixture of intact molecules (either uncut or repaired precisely), unrepaired DSBs and products of error-prone repair. Full negative-control time-course were collected for each target, consisting of 14 samples without sgRNA, accounting for DSBs occurring due to DNA degradation or other artifacts (Fig. 2D–F, for data see Source Data).

Both unrepaired DSBs and indel-containing products of error-prone NHEJ can be detected at all three targets, with indel-accumulation peaking between 48 and 72 h (Fig. 2A–C). DSBs first appear early in the time course (Fig. 2A–C), with significant evidence at 6 h in all 3 targets, compared to the controls (Fig. 2D–F). This suggests that RNPs are rapidly trafficked to the nucleus after transformation. Consistent with a longer repair process, molecules containing evidence of repair-error take hours to accumulate, with indels first detectable at low amounts at the 6 h time-point and increasing following 24 and 36 h (Fig. 2A–C). Out of the three targets analyzed, *PhyB2* displays both the highest frequency of indel products of repair-error and quantity of DSBs (Fig. 2C). While accumulating far fewer indels than *PhyB2, Psy1* has many detectable DSBs, reaching nearly 12% of the pool by 72 h, approximately equal to the quantity of edited (indel) molecules (Fig. 2A). These results suggest that efficiency of SpCas9 DSB induction at this locus may be underestimated by examining indels alone.

Further insight into the dynamics of this process can be gained by evaluating the kinetics of the type of indels emerging from error-prone repair along the time-course. To this end, indels were categorized as either deletions, insertions, or deletions associated with 2 or more base pairs of flanking microhomology (Fig. S2). At the three targets, deletions rise rapidly (Fig. S2A–F), particularly at *CRTISO* and *PhyB2* (Fig. S2E, F). While, in contrast, insertions peak between 6-12 h, before

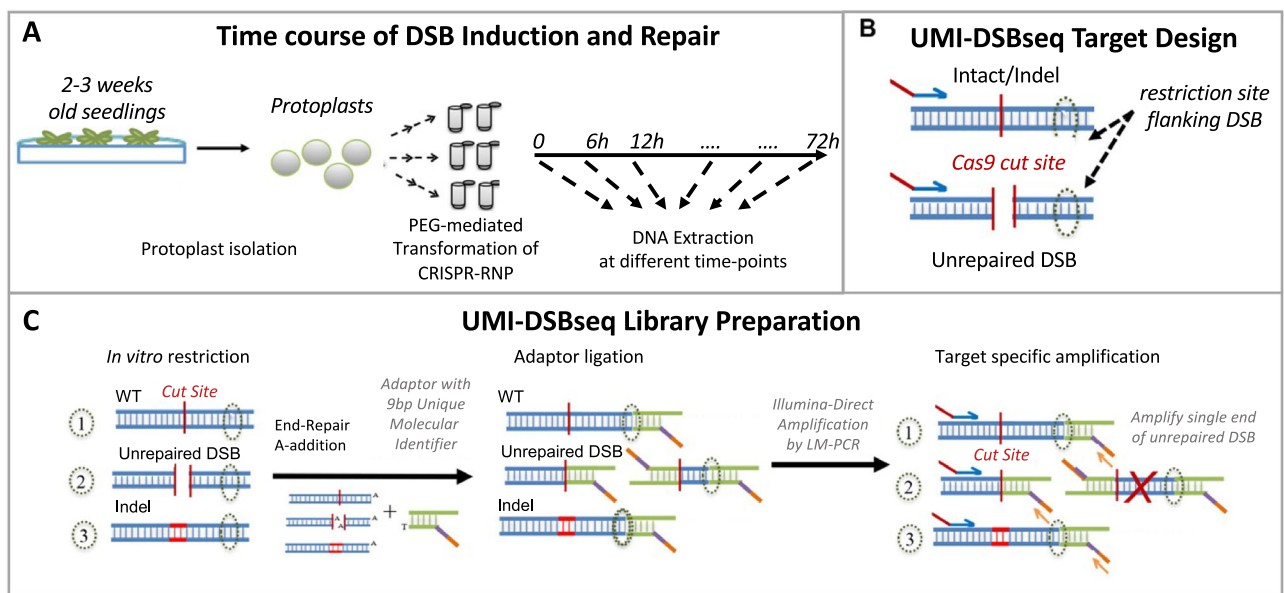

**Fig. 1 | UMI-DSBseq quantitative single-molecule sequencing of DSBs and repair products at three targets in tomato. A** Collection of time-course: mesophyll cell protoplasts are isolated from 2 to 3 week-old seedlings of M82 *Solanum lycopersicum*. Duplicate samples are prepared with 200,000 protoplasts for each of the 7 time-points along 72 h. CRISPR RNPs are introduced by PEG-mediated transformation. Samples are frozen at 0, 6, 12, 24, 36, 48, and 72 h after RNP introduction and DNA is extracted. **B** UMI-DSBseq target design: a primer specific to the target sequence (blue arrow), is coupled with a restriction enzyme site flanking the sgRNA target sequence to create an available end on intact molecules (WT or Indel) for ligation of adaptors. **C** UMI-DSBseq library preparation: DNA extraction from time-course collection, containing WT (1), unrepaired DSBs (2), and intact molecules containing indels (3), is restricted in vitro with the restriction enzyme identified flanking the target cut site (dashed oval). Following end-repair by fill-in and A-addition, Y-shaped adaptors composed of P7 Illumina flow-cell sequences and containing i7 indexes (orange) and 9 bp unique molecular identifiers (UMIs) in purple, are ligated to the unrepaired DSBs and restricted ends. Target-specific amplification by ligation-mediated PCR follows, with one primer identical to the adaptor sequence and containing the P7 Illumina tail (orange arrow) and one primer specific to the target sequence (blue arrow) with the P5 Illumina tail (red). This results in the amplification of a single end of the DSB between the SpCas9 cut site and the primer. The red X represents the non-captured end of the DSB.

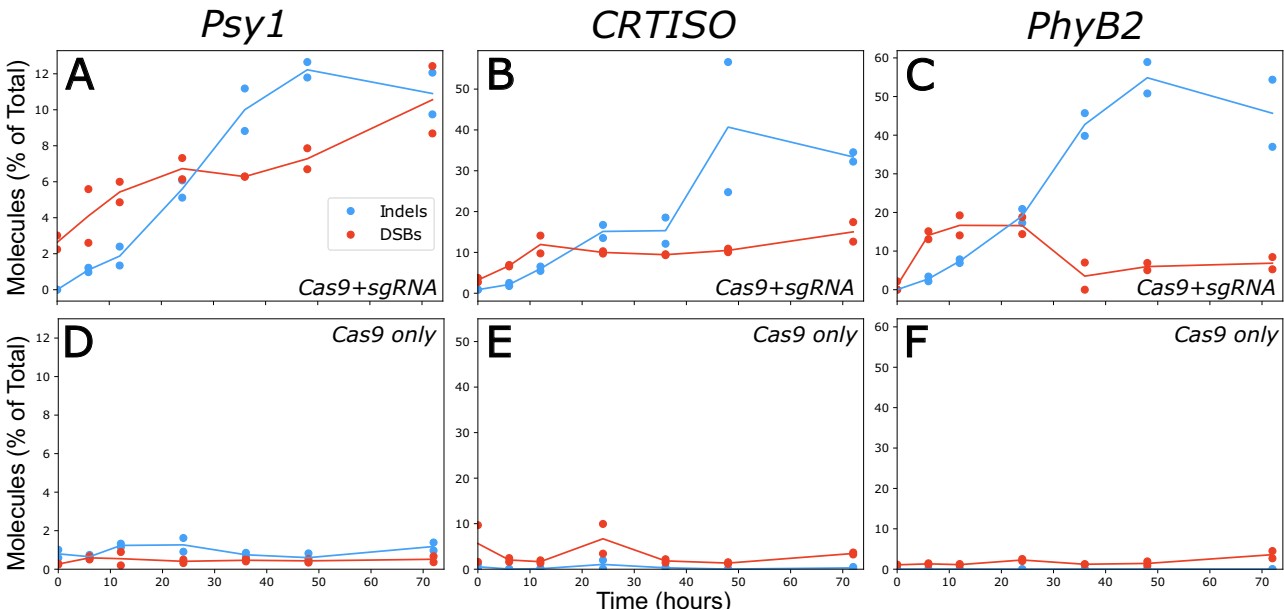

**Fig. 2 | Patterns of error-prone repair along 72 h time-course. A–F** Percent of molecules identified as unrepaired DSBs (red) and NHEJ-mediated indel (blue) out of total consensus sequences along 72 h, **A–C** experimental and **D–F** control time-courses for (**A**, **D**) *Psy1*, (**B**, **E**) *CRTISO*, and (**C**, **F**) *PhyB2* with dots representing percent of molecules from each replicate and a line representing the mean of the duplicates (see related: Fig. S2).

decreasing in total proportion, with this intriguing pattern most evident for *Psy1* (Fig. S2D). Microhomology-associated deletions were far less abundant at the examined targets, emerging slightly later. While present in rather negligible amounts in *CRTISO* and *PhyB2*, these indels are more abundant in *Psy1*, likely due to more microhomologies flanking this locus (Fig. S2D–F, Supplementary Data 1). At *Psy1*, there are two large microhomologies of 4 and 5 bp, TGTT and CCTTGT, overlapping and flanking the DSB cut site, respectively. While footprints associated with microhomology-mediated deletions resulting from the 5 bp sequence were not found in the data, 1–3% of all

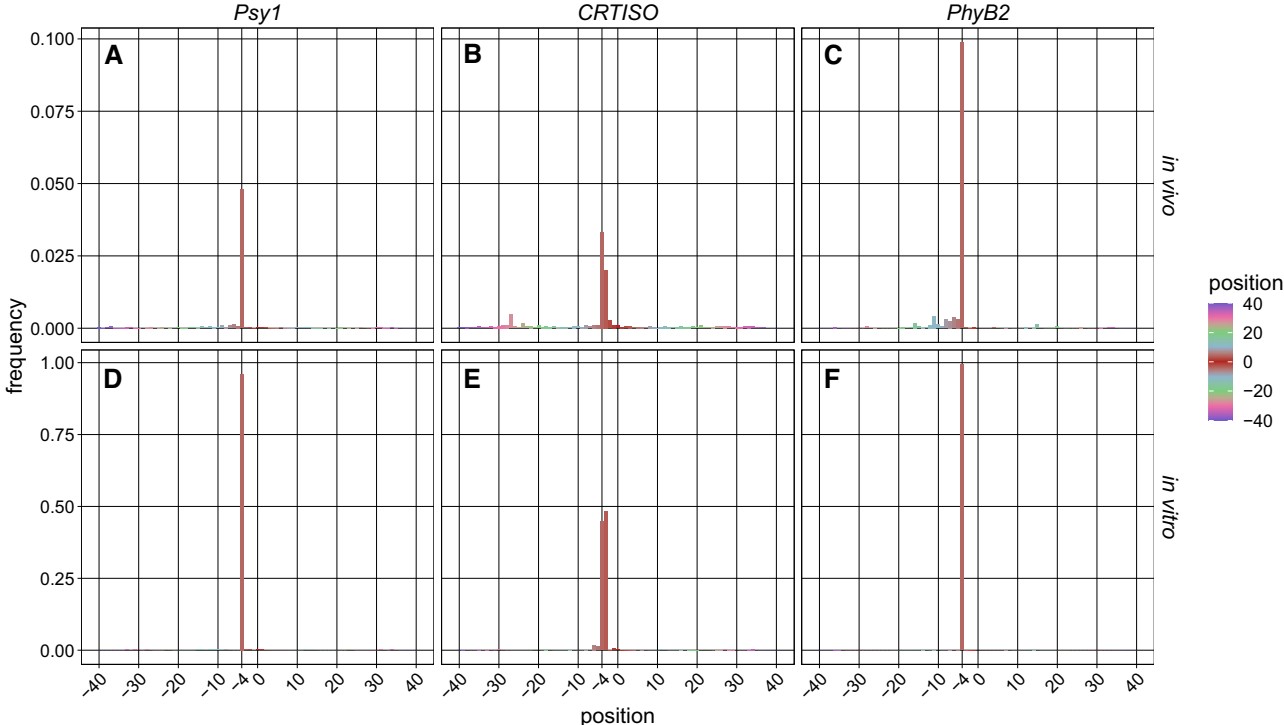

**Fig. 3 | Positions of DSB. A–C** Proportion of total DSBs (normalized to total molecules obtained in the in vivo time course) plotted by position along the target sequence with the expected cut site between -3 and −4 bp upstream of the PAM site, (**A**) *Psy1*, (**B**) *CRTISO* and (**C**) *PhyB2*. Bars indicating the proportion of DSBs captured at different positions over all time points of the time courses (*n* = 14) are shown with different colours based on the position, as shown in the legend. The positions of DSB for individual time points of the time courses are shown in Fig.S3. **D-E** Proportion of DSBs induced in vitro (normalized to total DSBs) plotted by the position of the captured end at (**D**) Psy1, (**E**) CRTISO, and (**F**) *PhyB2; (n* = 4).

deletions at this locus are the "TGTTGCCT" associated with the 4 bp microhomology (Fig. S2D, Source Data). A comparison of the results shown here (Fig. S2J–L), to a previous analysis of indels obtained *in planta* with the same gRNAs at *Psy1*[27], or at *CRTISO*[45], shows that most indels are similar. This suggests that the profile of indels is quite consistent, regardless of the cell types, and that the protoplast system is a good model for studying DSB repair in whole plants.

## Dual cleavage at *CRTISO*

At all three targets, the majority of DSBs are positioned around the expected SpCas9 cut site, between 3rd and 4th bps upstream of the PAM (Fig. 3A–C). Alongside the presence of negligible DSBs in the negative control time-courses (Fig. 2D–F) this evidence suggests that the UMI-DSBseq assay successfully captures Cas9-induced DSBs at the target sites. At *Psy1* and *PhyB2*, a sharp, single peak representing the vast majority of DSBs can be seen at the expected SpCas9 −3 position, along with a small number of processed DSBs captured in close proximity to the target site (Fig. 3A, C, Fig. S3). In contrast, at *CRTISO*, two primary DSBs arise, with most captured ends positioned at the −4 position from the PAM (Fig. 3B).

To investigate whether these DSBs result from CRISPR-Cas9-mediated cleavage or from end-processing during repair, DSBs were induced in genomic DNA in vitro (Fig. 3D–F) and compared to those characterized in vivo in the experimental time-course (Fig. 3A–C). While at *Psy1* and *PhyB2 in vitro*-induced DSBs occur precisely at the expected site, those captured at CRTISO, similarly to the time-course results, appear to contain two peaks (Fig. 3E, Fig. S3). The two main peaks are the same in the time course and in the in vitro data, with differences in their relative abundance (Fig. 3B, E). This supports that the two main −3 and −4 DBS termini are not due to in vivo end-processing but rather to dual cleavage by Cas9 at this locus.

## The kinetic model reveals efficient DSB induction coupled with precise repair

With the high-resolution data gathered, the rate of precise repair can be estimated using a set of ordinary differential equations from the broken (DSB) and precisely repaired (intact) state, describing the curves of the different states over time directly from the observed counts. We first implemented a simple 3-state model, with a similar structure to that previously reported by Brinkmann et al.[44], in which molecules are primarily classified as intact, DSB, and indels (Fig. 4). The rates of the flux between these different types of molecules are estimated as DSB induction ($K_{cut}$), repair-error ($E_{repair}$) and precise repair ($P_{repair}$) rates, respectively representing the flux from intact to DSB, from DSB to indel, and from DSB back to intact (wild-type), in terms of proportion of molecules per hour (Fig. 4A).

Evaluating the flux through the intermediate, unrepaired DSB state, which is directly observed, facilitates decoupling the effect of induction efficiency from repair fidelity at a given locus, compared to previous approaches[44]. Using direct measurements of the state of each sequenced molecule in the pool (Fig. 2, Fig. S2, Source Data), we implement a maximum likelihood estimation of the model's rates (Fig. 4, Table 1). This likelihood is maximized through extensive numerical optimization and uncertainty in the rate estimates is quantified through stratified bootstrap. Using this approach, the proportion of molecules flowing between the broken and repaired states can be estimated directly from the observations of single-molecule counts, revealing the rate of precise repair.

The fit of the model suggests that it offers a fair description of the process at the three targets along the time course, with most observations falling within the range of values predicted by the model (Fig. 4B–D, Table 1). To accommodate the fact that RNPs must enter the cell and act on the DNA, and that their activity can decay over time, we modeled induction as a time-dependent process, specifically as a

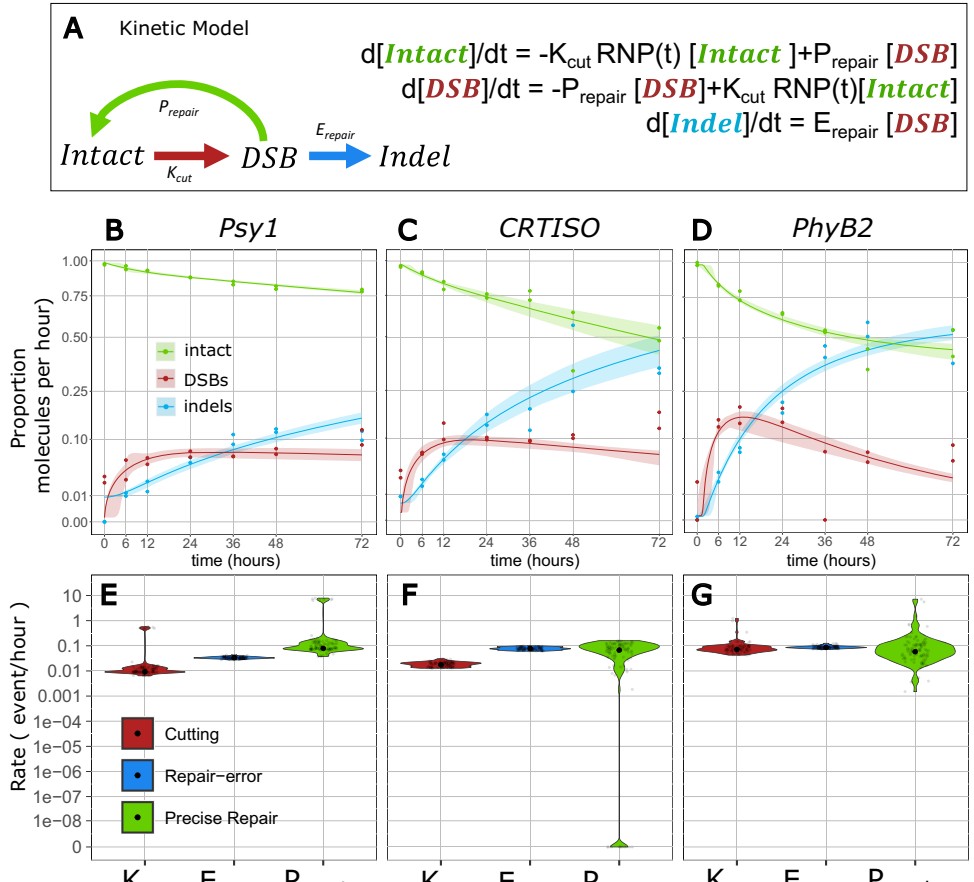

**Fig. 4 | A 3-state model of DSB induction and repair. A** schematic and equations for a 3-state model of DSB induction $K_{cut}$ in red, precise repair, $P_{repair}$ in green and Error-prone repair $E_{repair}$, in blue. **B–D** Predicted fit of the model (lines) at *Psy1* (**B**), *CRTISO* (**C**), and *PhyB2* (**D**) for Intact molecules (green), DSB (red), and indels (blue). Confidence intervals are shown as shadings and calculated from 100 bootstraps of the data. Observed data are represented as dots. **E–G** Rate constants estimated at *Psy1* (**E**), *CRTISO* (**F**), and *PhyB2* (**G**) in terms of number of events per hour per molecule. The smoothed distribution of the estimates obtained through the bootstrap procedure is shown as a violin plot. Mean estimate is shown as a black point. Grey points represent 100 instances of stratified bootstrap. (see related Table 1, Fig. S2, Table S1).

---

logistic increase in activity of RNPs with slope *r* and a maximal proportion of active RNPs 1-U, and with decay over time (rate *d*), as described in the Method section. Instead of imposing an induction curve measured a priori by a proxy, which might not capture the exact level of induction in the experiment, our kinetic model directly estimates the induction curve from the data at each individual time course, allowing for variations in genome search time and offering an approach more robust to mis-specifications of the induction curve. Simulations show that this approach is able to estimate accurately the repair dynamics of DSBs, even when an induction curve is not known a priori (Fig. S4). This reveals comparable patterns of induction across targets, with a rapid induction within the early hours of the time course and little decay within the 72 h sampling window, consistent with the presence of DSBs in the observed time courses (Fig. 2, Fig. S5, Table 1).

This is supported by a lack of cell division throughout the time course, as evidenced by fluorescent-assisted cell sorting (FACS) analysis of protoplast nuclei (Fig. S6), as well as by analysis of transformation efficiency by FACS of nuclei extracted from protoplasts transformed with RNPs of Cas9 fused to GFP (Fig. S7). The data confirm the rapid introduction of the RNPs to the nuclei starting at our initial sampling point, with most of the cells still containing detectable amounts of RNPs at 48 h post-transformation (Fig. S7). Note, however, that our initial point (here labelled 0 h) occurs only ~30 min after the transformation due to the time necessary to handle the samples. Thus, to account for this very rapid induction, we tested whether considering

this 30-min handling time—hence setting the initial time 0 h at the time of transformation, affects our results. All our rate constant estimates, including those of all models reported below, are minimally and non-significantly affected by including or not the 30 min transformation time (Fig. S8, Table S1). Thus, we report the results with no delay throughout the manuscript, while estimates obtained by including the transformation time are reported for all models in the Supplementary Information. The proportion of detectable RNP-containing cells, as estimated by FACS, can also be used as a proxy for the induction curve, under the assumption that this is the main limiting factor for DSB formation. Thus, we fitted an induction curve from FACS data using a binomial model and the parametrization described above (Fig.S9), and we repeated our estimates by imposing this curve to the model to constrain it and test its robustness. Estimates constrained with FACS data are highly similar to those in which the induction curve and the repair dynamics are co-estimated using UMI-DSBseq data, supporting the robustness of our estimates (Table S2, Fig.S10). This also indicates that it is possible to estimate the dynamics of repair without imposing a pre-determined induction curve, which might otherwise skew estimates if mis-specified.

The estimated rate of DSB induction suggests that molecules are broken at a rate of ~1–8% per hour, with rate constant $K_{cut}$ between 0.009–0.0718 of intact molecules per hour (Fig. 4E–G, Table 1). These estimates indicate that in total, 57.5%, 88.2% and 91.1% of all molecules were successfully cleaved within 72 h at *Psy1*, *CRTISO* and *PhyB2*,

**Table 1 | Rates and flow estimates for the 3-state model of DSB Repair for the 72 h time course**

| Target | Process | Rate[a] (proportion of total event/hour) | CI [1–99%][c] | Flow[b] after 72 h (proportion of total molecules) | CI [1–99%][c] | P-value |
|---|---|---|---|---|---|---|
| Psy1 | *Cutting* | 0.0092 | 0.0064-0.5434 | 0.5755 | 0.4031–31.0895 | <0.01 |
| | *Repair-error* | 0.0335 | 0.0276-0.0419 | 0.1518 | 0.1364-0.1683 | <0.01 |
| | *Precise Repair* | 0.0791 | 0.0373-7.2711 | 0.3581 | 0.1812-30.6583 | <0.01 |
| | e (processed DSB) | 0.2052 | 0.1659-0.2392 | / | / | <0.01 |
| | U (uncut fraction) | 0 | 0–0.0001 | / | / | 0.98 |
| | r (induction speed) | 9402.089 | 2.5443-20909.0328 | / | / | <0.01 |
| | d (induction decay) | 0 | 0–0 | / | / | <0.01 |
| | Repair accuracy | / | / | 0.7024 | 0.5478-0.9951 | <0.01 |
| CRTISO | *Cutting* | 0.0175 | 0.0129-0.029 | 0.8815 | 0.5429-1.2973 | <0.01 |
| | *Repair-error* | 0.0777 | 0.0611-0.0976 | 0.4375 | 0.3604-0.5178 | <0.01 |
| | *Precise Repair* | 0.0675 | 0–0.1563 | 0.3799 | 0–0.8654 | 0.1 |
| | e (processed DSB) | 0.32 | 0.2332-0.3908 | / | / | <0.01 |
| | U (uncut fraction) | 0 | 0–0.1179 | / | / | 0.92 |
| | r (induction speed) | 48.8311 | 5.3069-24293.2738 | / | / | <0.01 |
| | d (induction decay) | 0 | 0–0.0048 | / | / | <0.01 |
| | Repair accuracy | / | / | 0.4648 | 0–0.7 | 0.1 |
| PhyB2 | *Cutting* | 0.0718 | 0.0434-0.9763 | 0.9113 | 0.5786-26.7611 | <0.01 |
| | *Repair-error* | 0.0864 | 0.0741-0.122 | 0.5282 | 0.4966-0.579 | <0.01 |
| | *Precise Repair* | 0.0583 | 0.0036-5.5634 | 0.3567 | 0.0229-26.0203 | <0.01 |
| | e (processed DSB) | 0.0817 | 0.048-0.1353 | / | / | <0.01 |
| | U (uncut fraction) | 0.4047 | 0–0.4395 | / | / | 0.04 |
| | r (induction speed) | 8.0297 | 2.9301-1079.26 | / | / | <0.01 |
| | d (induction decay) | 0 | 0–0.0073 | / | / | <0.01 |
| | Repair accuracy | / | / | 0.4031 | 0.0414-0.9784 | <0.01 |

[a]Rates are reported as the number of events per molecule per hour.
[b]The flow is reported as the proportion of molecules that experienced that event at the end of the experiment.
[c]Confidence intervals (CI) are reported as the 1% and 99% percentiles of the estimates obtained from 100 stratified bootstraps of the data, while p-values as the proportion of bootstraps with value smaller or equal than 0 (one-sided test). When none of the 100 bootstraps had value equal to 0 we reported *p*-values as <0.01. The induction curve is modeled as a logistic increase in activity of the RNPs with speed r, a fraction U of cells upon which the RNPs do not cut DNA, and a decay. *d* An error parameter. Describes the proportion of DSBs which show DSB ends not coinciding with the expected position (see Methods: Determining the parameters of the induction curve). Repair accuracy is computed as the proportion of DSBs repaired precisely over all repaired DSBs (precise and error repair) at the end of the time course.

respectively (Fig. 4E–G, Table 1). At the 3 targets, the rate of repair error, $E_{repair}$, is estimated between 0.034–0.086 of DSBs per hour (0.0335, 0.0777, 0.0864 for *Psy1*, *CRTISO*, and *PhyB2*, Fig. 4E–G, Table 1) leading to the prediction that 15.2%, 43.8% and 52.8.% of the molecules present in the sample are repaired as indels over the time-course (Fig. 4E–G, Table 1). This is consistent with the mean proportion of indels directly observed in the 48 and 72 h sampling points at the three targets (Fig. 2A–C).

Low indel accumulation coupled with relatively efficient cutting suggests the possibility that many of the broken molecules are repaired through high-fidelity mechanisms, leading to precise repair. Indeed, the rate of precise repair, $P_{repair}$, is estimated between 0.058-0.079 molecules per hour at the three targets, suggesting that 35.81%, 37.99%, and 35.67% of all molecules have been broken and repaired precisely following the 72 h time-course for *Psy1*, *CRTISO* and *PhyB2*, respectively (Fig. 4E–G, Table 1). Statistical significance of precise repair is confirmed by 100 stratified bootstraps of the data for both *Psy1* and *PhyB2* (p-value < 0.01, Table 1). While for *CRTISO* the point estimate of the rate of precise repair (0.068) is comparable to those of *Psy1* and *PhyB2* (0.058-0.079), the same bootstrap approach could not rule out that precise repair is absent at this target, with confidence intervals 0–0.156 precise repair events per hour per molecule (Fig. 4E–G, Table 1). However, a model comparison of the 3-state model versus a model in which repair is only error-prone was significantly preferred for the full dataset at all targets (AIC-based relative likelihood <10$^{-5}$), as well as stably supported at *Psy1* and *PhyB2* in the bootstrapped data (100% of the bootstraps of *Psy1* and 96% of *PhyB2*,

and 81% for *CRTISO*, Fig. S11, Table S3). Altogether, these results lead to the conclusion that DSB induction by the CRISPR-Cas9 RNPs is efficient; and is coupled with relatively high rates of precise repair.

### UMI-DSBseq captures the formation and processing of unrepaired DSBs

Captured DSBs may represent a pool of both newly formed breaks directly induced by Cas9 (hereafter defined as direct DSBs) as well as intermediates of active repair processes as they alter the DSB ends (hereafter defined as processed DSBs). Processed ends are the likely intermediates of indel formation upon repair, although in principle they can also be repaired precisely, e.g., through HR. In fact, the loss or gain of base-pairs at the ends resulting from errors in repair, would form transient intermediates offset from the expected position. Deletion intermediates would result in a shorter than expected captured DSB; while insertion intermediates (extended) or those of DNA synthesis (guide-side) would extend past the expected position (Fig. 5A). While DSBs captured at the targets are distributed around the expected break-site, processed DSBs were also detected at all targets while absent in the negative controls as well as in the in vitro samples (Fig. 3). To assess the dynamics of these putative intermediates, DSBs were categorized in direct or processed DSBs depending on their position relative to the expected cut site (Fig. 5).

Classifying DSBs in all three targets reveals that the majority of DSBs along the time course are direct DSBs (−3bp from the PAM site) with no evidence of bp loss or gain (Fig. 5), suggesting they may be good substrates for the precise repair predicted by the 3-state model

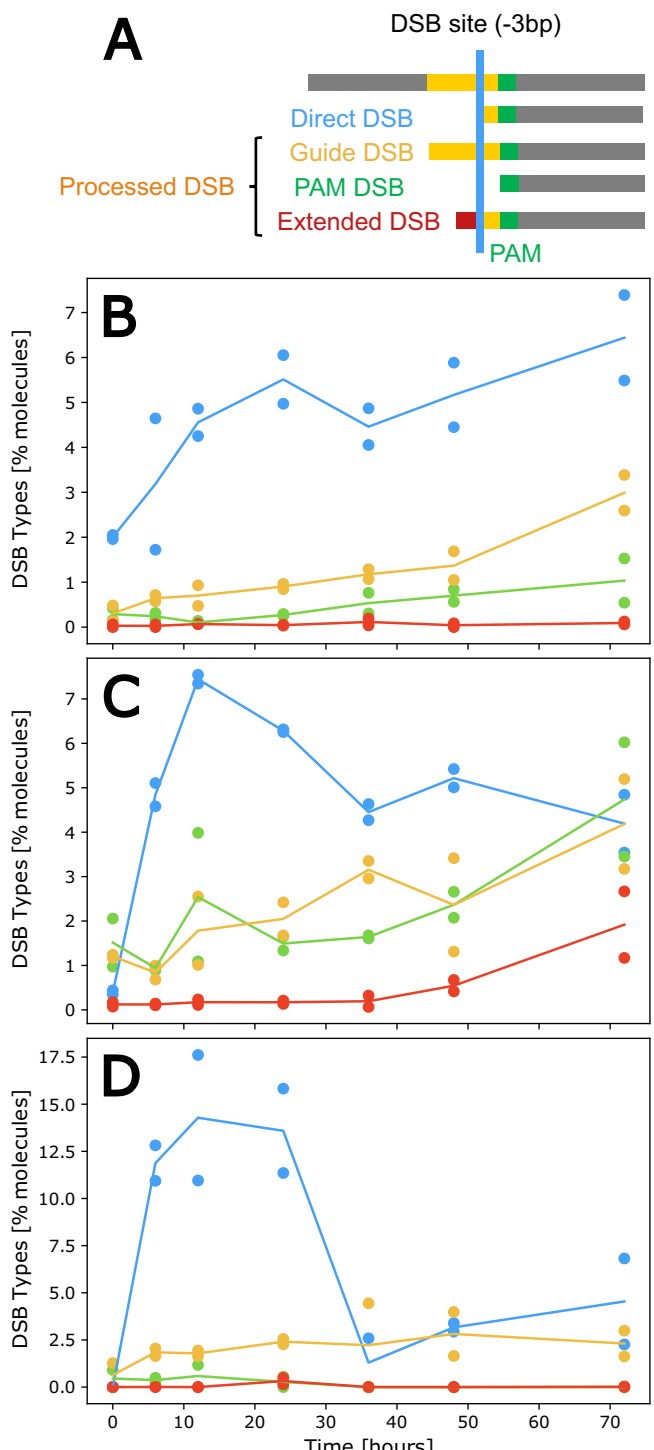

**Fig. 5 | Characterizing putative intermediates of DSB repair. A** Schematic of categories of DSB types: Direct DSB (−3 bp from the PAM site), Processed DSBs (putative repair intermediates) classified as Guide, PAM, and Extended DSBs. **B–D** Percent of DSB types along the time-course of repair, for *Psy1* (**B**), *CRTISO* (with perfect DSBs including both -3 and −4 bp) (**C**) and *PhyB2* (**D**). Direct DSBs are shown in blue, guide-side DSBs in yellow, PAM-side DSBs in green and extended DSBs with extensions that do not match the target sequence in red.

(Fig. 4, Table 1). However, at all 3 targets, some putatively processed DSBs can be identified, with classification revealing differing dynamics between the different types compared to the expected direct DSB (Fig. 5B–D). The curve of direct DSBs rises rapidly and begins depleting in both *PhyB2* and *CRTISO* (Fig. 5C, D). In contrast, at *Psy1*, the shapes of

direct and processed DSBs are similar, with continued increase in abundance as the time course progresses, perhaps suggesting evidence of continued cutting of precisely repaired molecules that have re-entered the pool (Fig. 5B).

At *CRTISO*, since two types of primary DSBs are observed and confirmed in the in vitro assay, in the absence of the molecular machinery for DNA repair, we infer that both types of DSBs can be induced directly by Cas9 and so can be categorized as direct DSBs (Fig. 3B, E). Note, however, that the observed proportions of the two primary DSBs differ noticeably in vivo and in vitro (higher −4 bp than −3 bp in vivo, higher −3 bp than −4 bp in vitro) so that we cannot exclude that repair processes affect the two differently.

**Processed DSBs represent transient intermediates of error-prone repair**

A dynamic model was built, including 'processed' DSBs as a fourth state of the molecules, generated from direct DSBs (Fig. 6, Fig. S12, Table S4; for the model with transformation time, see Fig. S13, Table S5; for the model using induction curves estimated via FACS, see Fig. S14, Table S6). In this model, the rates of the flux between these different types of molecules are estimated as DSB induction ($K_{cut}$, from intact to direct DSB), processing of direct DSBs ($K_{processing}$), and repair of both direct ($E_{direct}$, $P_{direct}$) and processed DSBs ($E_{processed}$, $P_{processed}$) by error-prone and precise mechanisms. Using this approach, we can evaluate the contribution of DSB processing to high-fidelity and error-prone repair, estimating the flux through the intermediate state prior to re-ligation (Fig. 6). Altogether, the fit of the 4-state model makes predictions compatible with those estimated by the 3-state, with comparable values, and largely overlapping confidence intervals (Fig. 6, Table 1 versus Table S4). However, the 4-state model is significantly preferable (AIC-based relative likelihood $< 10^{-5}$) over the 3-state model on the full dataset and in all 100 bootstraps of the data, at all 3 targets, suggesting it provides a better description of the complex dynamical process of DSB repair (Fig. S12G–I). Using this model, the rate of DSB processing and the type of repair at processed DSB intermediates can be estimated. Simulations show that, despite the larger number of parameters, the 4-state model is accurate and shows low false positives (Fig. S15).

Consistent with their identification in the time-course data, processing of direct Cas9-induced DSBs is detected at all targets, accounting for 3–12% of total molecules along the time-course, with $K_{processing}$ estimated between 0.009–0.0224 of direct DSB molecules per hour (Fig. 6B–D, Table S4). For *CRTISO*, two direct DSBs (at positions −3 and −4) can be seen both in vitro and in the time-course data (Fig. 3G, H) and it is possible that each could display different kinetics. For instance, if one of the DSBs were staggered, as suggested in previous works[47,48], such a break might produce a good substrate for precise re-ligation. Thus, we included an additional model in which we considered the −4 bp cleavage as a direct end rather than processed (Fig. S16). While support for precise repair at this locus is still limited, this model results in a higher point estimate for precise repair (Fig. 6E, F), indicating that if precise repair occurs at this locus, it likely comes from DSB formed directly at −4 (dual cleavage site), supporting the hypothesis that it may represent a staggered cut formed by SpCas9 (Fig. 6 E, F, Fig. S16). Nevertheless, we cannot rule out that some of the −4 ends seen in vivo are the result of processing, in which case precise repair would be negligible (Fig. 6 F).

Precise repair of processed DSB ends appears negligible at *Psy1* and *PhyB2*, with rate constant $P_{processed}$ estimated at 0 (Fig. 6B, D, Table S4). At *Psy1*, only a small fraction of processed DSB can be detected in general, and they remain largely unrepaired (Fig. 6B, G, Table S4). In contrast, processed intermediates at *PhyB2*, accounting for 12% of total molecules in the pool following 72 h, are predicted to be repaired with error, at a rate of 0.1679 per hour (Fig. 6D, H, Table S4). These estimates suggest that all processed DSBs at this target represent transient intermediates of error-prone repair. Altogether, the high-resolution

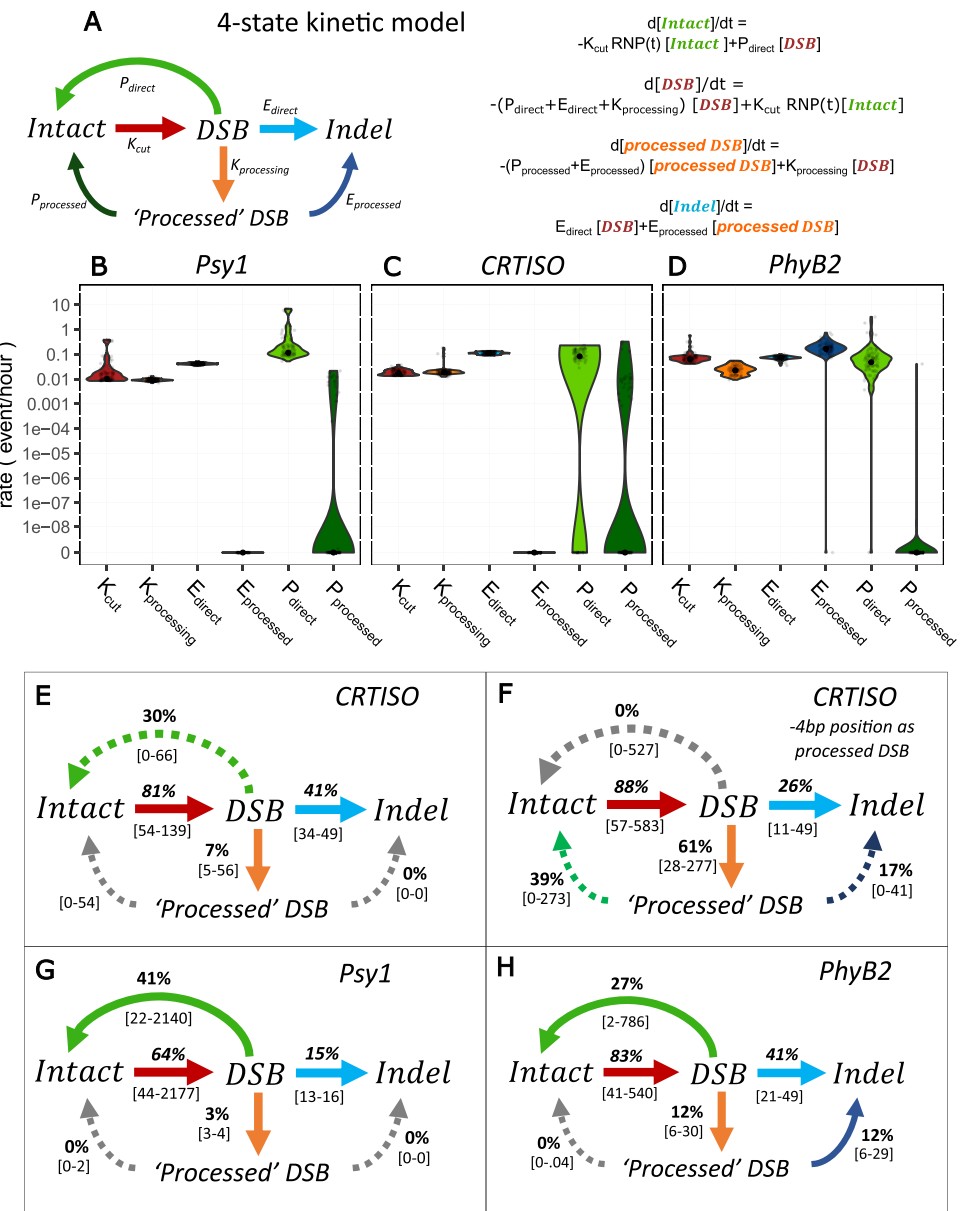

**Fig. 6 | A 4-state model of DSB induction and repair, including processed ends.**
**A** schematic and equations for a 4-state model of DSB induction, $K_{cut}$, ends processing, $K_{processing}$, repair from directly induced DSBs, namely precise repair, $P_{direct}$ and error repair $E_{direct}$, and from putative repair intermediates 'processed' DSB, including precise repair $P_{processed}$ and error repair $E_{processed}$. **B–D** Violin plots of rate constants estimated at *Psy1* (**B**), *CRTISO* (**C**), and *PhyB2* (**D**) in terms of proportion per hour (as in Fig.4E–G). **E–G** Schematic representation of the dynamic flow estimated in terms of percent of total molecules following 72 h, at *CRTISO* when (**E**) the −4 position DSB is considered as a directly induced (DSB) or (**F**) as a 'processed' DSB, (**G**) *Psy1*, and (**H**) *PhyB2*. Grey arrows indicate rates estimated as 0, dashed arrows represent confidence intervals overlapping 0. CIs indicated in grey brackets and calculated from 100 iterations of the bootstrap. See also Table S4, Fig. S12.

characterization of DSB intermediates facilitated by the UMI-DSBseq, coupled with a 4-state kinetic model, further supports estimated rates of prominent precise repair while revealing ambiguous dynamics associated with repair-error, at *CRTISO* and *PhyB2*.

**Estimates of the precise repair of DSBs in a 24 h high-resolution time courses**
While we explored DNA repair over time courses stretching 72 h and accounted for the reduction in RNPs activity by inferring its induction curves, it is possible that changes in the cell cycle, RNP and cell viability could affect our estimates. To address these caveats, we generated additional time courses spanning a shorter time (24 h) and at higher time resolution (with time points at 2,4,6,12, and 24 h) for all targets (Fig. 7). Analyzing these time-courses with the 3-states model, we

confirm the presence of precise repair of DSBs at *Psy1* (*p*-value < 0.01) and *PhyB2* (*p*-value = 0.01), while precise repair of DSBs is not significantly higher than 0 for CRTISO (*p*-value = 0.93) (Fig. 7, Table 2, Table S7 and Fig. S17 for the model with transformation time). Rate estimates for precise repair do not differ significantly between the 24 and 72 h time courses for the three loci (Table 1, Table 2). Total precise repair of DSBs at *Psy1* is also supported (*p*-value < 0.01) in the 4-state model (Table S8 and Fig. S18 for the model with transformation time). Interestingly, the high-resolution time course at *Psy1* supported the repair of processed DSBs (*p*-value = 0.02), while this was not significant in the 72 h time course. Note that for the 24 h time course, the 4-state model generally receives less support compared to the 72 h time course: while for *Psy1* the 4-state model provides a statistical improvement over the 3-state model in both time courses (*p*-value =

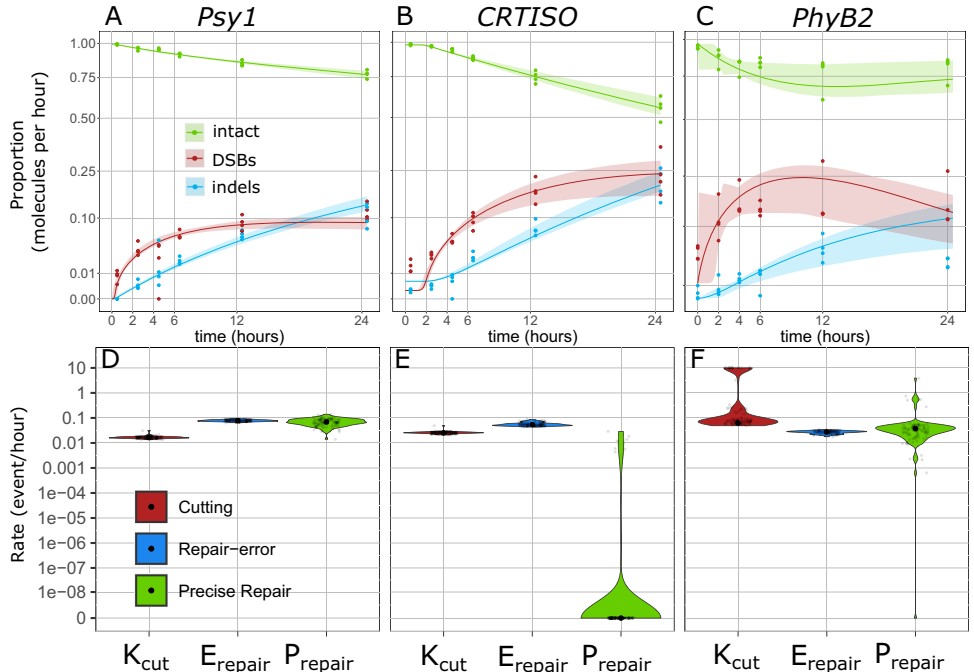

**Fig. 7 | High resolution time courses over 24 h. A–C** Fit of the 3-state model to the data and respective estimates of the rate constants (**D**, **E**) for *Psy1* (**A**, **D**), *CRTISO* (**B**, **E**), *PhyB2* (**C**, **F**). Mean trajectory and rate estimates are shown as continuous lines (**A–C**) and black dots (**D–F**), while the distribution of 100 bootstraps is shown in lighter colours.

**Table 2 | Rates and flow estimates for the 3-state model of DSB Repair for the 24 h high-resolution time course**

| Target | Process | Rate (proportion of total event/hour) | CI [1–99%] | Flow after 24 h (proportion of total molecules) | CI [1–99%] | *P*-value |
|---|---|---|---|---|---|---|
| Psy1 | Cutting | 0.0164 | 0.0139-0.0217 | 0.3439 | 0.2828-0.4287 | <0.01 |
| | Repair-error | 0.0777 | 0.0666-0.0929 | 0.1348 | 0.1193-0.1555 | <0.01 |
| | Precise Repair | 0.0691 | 0.028-0.131 | 0.1198 | 0.0491-0.2207 | <0.01 |
| | *e* (processed DSB) | 0.1228 | 0.1012-0.1421 | / | / | <0.01 |
| | *U* (uncut fraction) | 0 | 0–0.0031 | / | / | 0.97 |
| | *r* (induction speed) | 12336.24 | 116.261-22425.4293 | / | / | 0.01 |
| | *d* (induction decay) | 0 | 0–0 | / | / | <0.01 |
| | Repair accuracy | / | / | 0.4706 | 0.2336-0.6193 | <0.01 |
| CRTISO | Cutting | 0.0251 | 0.021-0.0299 | 0.4339 | 0.3808-0.4795 | <0.01 |
| | Repair-error | 0.0538 | 0.0445-0.082 | 0.197 | 0.1767-0.2457 | <0.01 |
| | Precise Repair | 0 | 0–0.0187 | 0 | 0–0.0598 | 0.93 |
| | *e* (processed DSB) | 0.1016 | 0.0779-0.1133 | / | / | <0.01 |
| | *U* (uncut fraction) | 0 | 0–0.0043 | / | / | 0.97 |
| | *r* (induction speed) | 10.1122 | 8.1819-11.9085 | / | / | <0.01 |
| | *d* (induction decay) | 0 | 0–0.0176 | / | / | <0.01 |
| | Repair accuracy | / | / | 0 | 0–0.2371 | 0.93 |
| PhyB2 | Cutting | 0.0624 | 0.0537-10 | 0.41 | 0.1718-194.3107 | <0.01 |
| | Repair-error | 0.0274 | 0.0191-0.0334 | 0.1195 | 0.0568-0.1541 | <0.01 |
| | Precise Repair | 0.0372 | 6e-04-55.8065 | 0.162 | 0.0021-181.6375 | 0.01 |
| | *e* (processed DSB) | 0.1455 | 0.1238-0.1645 | / | / | <0.01 |
| | *U* (uncut fraction) | 0 | 0–0.3821 | / | / | 0.83 |
| | *r* (induction speed) | 7838.409 | 0-33839.845 | / | / | 0.08 |
| | *d* (induction decay) | 0.0922 | 0–1.375 | / | / | <0.01 |
| | Repair accuracy | / | / | 0.5754 | 0.0272-0.9996 | 0.01 |

Description as in Table 1.

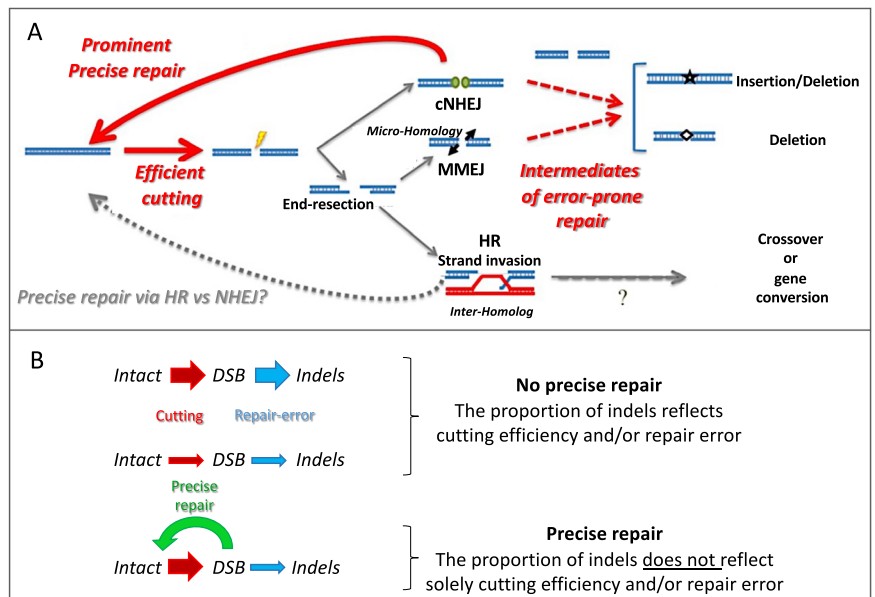

**Fig. 8 | UMI-DSBseq coupled with likelihood-based modeling reveals variables in the processes of DSB induction and repair. A** Decoupling of induction and repair revealing dynamics characterized by relatively efficient cutting and prominent precise repair. **B** Model of DSB repair in the absence of precise repair was rejected by AIC relative likelihood. Models of DSB repair in the presence of precise repair can explain the rate of indels formation at various loci (3-state is shown here for simplicity, but the same conclusion applies to the 4-state model). The thickness of the arrows indicates the efficiency, i.e., thicker arrows have a higher proportion of molecules.

0.04), for *CRTISO* (*p*-value = 0.42) and *PhyB2* (*p*-value = 0.29) the 4-state model is not significantly supported in the 24 h one (Fig. S13, Table S8, with transformation time in Fig. S19, Table S9). Results are similar when using FACS to estimate the induction curve (Fig. S20, S21, Table S10–S11). The only perceivable difference is that estimates of precise repair for *PhyB2* are even higher and show smaller confidence intervals for *PhyB2* using the FACS-estimated induction curve, both in the 3-state and in the 4-state model (Fig. S20, S21).

## Discussion

In this work we studied the kinetics of the DSB repair process using a molecular and computational toolkit, UMI-DSBseq. Starting with measurements of unrepaired DSBs and repair products, we characterized the dynamics of the repair process through kinetic modeling, successfully decoupling cutting efficiency from repair fidelity and revealing aspects of both high-fidelity and error-prone repair. These results establish that precise repair is a prominent feature of CRISPR-Cas9-induced DSB repair in somatic plant cells and provide insight into the contribution of processed-ends to indel formation. The finding that precise repair occurs at an important fraction of DSB is in line with the evidence provided by the analysis of junctions in large deletions or inversions induced by double cuts, which show repair without any nucleotide gain or loss[49]. Together, the results shown here have numerous implications in both biotechnology and the study of cellular processes involved in genome stability. Applying UMI-DSBseq to a large number of loci may help optimize the design of gRNAs. For example, being able to predict the rates of DSB induction, error-prone or precise repair would be very useful: a target with high DSB induction but low rate of precise repair and high error-prone repair may predict effective mutagenesis; while a strong DSB induction but slow repair rate (precise and error-prone) might be preferable in HDR experiments.

Deconvoluting DSB induction and repair fidelity requires inferring the elusive flux from broken to intact molecules. This was achieved through direct measurements of intact, broken, and repaired molecules in a single assay. Compared to similar methods based on LM-PCR[44,50–55], UMI-DSBseq facilitates simultaneous capturing of DSB intermediates alongside repair products, resulting in high-resolution quantification and characterization of all molecules along the time-course of repair. Moreover, the inclusion of UMIs enables a quantitative characterization of each repair intermediate and product through single-molecule sequencing. This has the advantage of allowing more direct measures of repair processes and to employ likelihood based statistical tools that can quantify and reduce our uncertainty. With insight into the high-resolution temporal dynamics at various loci, both from characterizing the unrepaired ends and probing repair kinetics, it can help us to better understand the factors impacting the type of pathway and errors that may emerge during repair. The UMI-DSBseq approach could be applied to a wide range of genome stability studies, such as the analysis of hotspots of recombination or transposition events, across a wide range of cell types.

UMI-DSBseq allows to observe the dynamics of different types of DSBs, allowing in principle the investigation of how different types of breaks get repaired over time. We analyzed the results using a 3 or 4-state model, the latter discriminating between DSBs at the expected cut sites and other DSBs. The 4-state model leads to the same conclusions as the 3-state model, namely, that about 40–70% of the DSBs repair events are repaired precisely (See Repair accuracy in Table 1, defined as the ratio of precise DSB repair out of total DSB repair, precisely or with indels). However, the 4-state model adds a new dimension, facilitating exploration of complex dynamics and capturing the transition state following DSB formation, as in *CRTISO* (Fig. 6E, F, Fig. S11, Fig. S12, Fig. S13) or as with *PhyB2*, where processing of the ends contributes to error-prone repair (Fig. 6H). Upon repeating the time-courses with denser sampling (Fig. 7) we can see that efficient DSB induction and precise repair remain prominent and robust features at both *Psy1* and *PhyB2* regardless if the time course was 24 or 72 h, while in both cases, *CRTISO* did not show evidence of precise repair (though results for this target might ambiguous because of two adjacent cuts). For the targets investigated here, our results indicate that repair fidelity is an important determinant of the proportion of indels arising following a DSB. Nevertheless, it may vary along the genome, for instance, due to differences in chromatin contexts; this highlights that DSB induction is likely underestimated when evaluated exclusively based on the products of error-prone repair, i.e., indels (Fig. 8).

The system presented here will be important for studying DSB repair. For example, the contribution of precise repair to the DSB repair process has been elusive so far. It might occur via precise NHEJ of unprocessed ends. In addition, precise repair could be achieved through HR, between sister chromatids or homologues. Recent works show evidence of the role of inter-homologue HR in DSB repair in somatic cells; however, the rates have been shown to be very low[27,28,30,56]. In our experimental system, the majority of protoplasts were characterized as being in the G1 stage of the cell cycle (see Fig. S5), supporting that repair is likely by NHEJ rather than by HR with a nearby sister chromatid. Validating repair kinetics in NHEJ or HR mutants or using DNA repair inhibitors would enable to further dissect the mechanism of precise repair. Likewise, the effect of genes affecting end processing, either through deletion or addition of nucleotides, could be assessed using the 4-state model. These genes may contribute to error-prone repair or hinder the repair process leading to unrepaired ends and chromosomal rearrangements[57,58].

Despite the widespread use of CRISPR/Cas technology, the ability to faithfully predict the proportion of indels following DSB induction remains challenging[40,59–61]. One likely reason for that is that these studies have not decoupled the effect of cleavage from that of repair. In the absence of precise repair, the main factor limiting indels formation at a given locus would be the cleavage efficiency of the targeted endonuclease, namely the sgRNA and Cas9 protein (Fig. 8B). If we were to evaluate the cleavage efficiency based on indel accumulation alone, we might conclude, for example, that *Psy1* is an inefficient target, showing a 15% error repair while the DNA cleavage was ~60%, namely four times higher than the repair error (Table1, Fig.6 G). Hence, we would have underestimated by ~4 times the cleavage efficiency of this target. This is mostly caused by a large fraction of DSBs that are precisely repaired, specifically 36–41%, Table 1, Fig. 6G. Though we notice some variability between different time courses and targets, our results show that precise repair plays an important role in shaping the frequency of indels upon CRISPR/Cas9 treatment, which is often used as a measure of editing efficiency.

Besides indel formation, neglecting precise repair would have also affected our evaluation of the speed of Cas9 digestion. For example, in the case of *Psy1*, we estimate a rate of 0.0092 cuts per hour (Table 1), corresponding to an average time before a DNA molecule is cleaved of 108.7 h. Without precise repair, the cutting rates would be 0.0205 (Table S2), and the average time is 48.8 h, underestimating the cleavage rate by >2-fold. Cleavage efficiency might also be affected by other parameters such as the speed of DSB induction[62] and Cas9 residence time[63]. Therefore, our results suggest that the efficiency of indels induction may vary due to the combined effect of cleavage efficiency, precise repair, ends processing, and error-prone repair (Fig. 8). Note that several factors might affect the efficiency of RNP complexes and the dynamics of DSB repair, not only across genomic sites but also across life cycle stages, stress conditions, cell types and physiological states. Such experimental variability possibly explains the small differences detected in the specific values of repair rates as seen between the 24 h and 72 h time courses. Despite these subtle variations, precise repair played a major role in DSB repair across experiments.

The protoplast system offers an opportunity to study the dynamics of DSB induction and repair within an enclosed and synchronized plant cell system. However, even though the mutational profile was similar in protoplasts and whole plants, the leaf mesophyll protoplast used here are non-dividing cells at the G1 stage, and CRISPR-Cas9 transfection was done using RNP rather than transgenic plants, which might narrow the scope of the conclusions. The effect of the cell state and type on repair fidelity remains to be investigated, with a possible role for HR in dividing cells. In addition, the selected targets used in this study were all three in exons, which could explain the similarity in cleavage efficiency (64-88% range) between the

targets. Future studies leveraging our approach will be able to clarify the dynamics of DSB repair in different chromatin contexts[64], in targets that display markedly different indels frequency[65,66]. Further application of the UMI-DSBseq system across different cell types and cell states and in different targets will lead to a better understanding of DSB induction efficiency and repair fidelity in general.

## Methods

### Experimental model and subject details

**Plant materials and growth.** M82 Tomato seeds were gas sterilized and sown on Nitsch media in magentas under long-day conditions, at 23 C.

Resources and reagents are listed in Supplementary Table 12.

### Method details

**Protoplast isolation and purification.** The protocol was adapted from Yoo et al.[67]. Briefly, first true leaves of 14 to 20-day-old seedlings are sliced into thin strips and immersed in 15 ml of enzyme solution (3.65 g Mannitol, 2 ml 0.5 M MES pH = 5.7, 500 ul 2 M KCl, 0.75 g Cellulase R10 Duchefa, 0.2 g Macerozyme Duchefa, 500 ul 1 M CaCl2, 0.05 g BSA) in a Petri dish, and incubated in the dark overnight with mild shaking (25 RPM) for 14–16 h. Extracted protoplasts strained through a 0.1 um filter and washed with W5 solution (154 mM NaCl, 125 mM CaCl2, 5 mM KCl, 2 mM MES pH = 5.7). Healthy, intact protoplasts are isolated using sucrose gradient (23%, 11.5 g per 50 ml). Following additional washing with W5, protoplasts are diluted to a concentration of 1 million cells/ml MMG solution (0.4 M Mannitol, 15 mM MgCl2, 4 mM MES pH = 5.7).

**sgRNA preparation.** sgRNAs are purchased from IDT as tracrRNA and target-specific crRNA. Both are diluted in TE buffer to a concentration of 200 uM each. Equimolar ratio of the two components are mixed with Nuclease Free Duplexing Buffer and heated at 95 C for 5 min before slowly cooling to room temperature. Stored at -20 C until day of transformation.

**SpCas9 protein expression.** SpCas9 protein was expressed from pET-28b-Cas9-His[68] (Addgene plasmid # 47327 RRID:Addgene_47327) by the proteomics Unit at the Weizmann Institute.

**Preassembly of CRISPR RNP.** For each sample 10 ug of SpCas9 and 20 ug of duplexed sgRNA are mixed with NEB Buffer 3.1 at 20 ul total volume and incubated at room temperature for 15 min. RNPs are prepared immediately prior to transformation.

**PEG-mediated transformation of CRISPR RNP.** 20 ul preassembled RNP is added to 200,000 protoplasts in 200 ul MMG solution. An equal volume of PEG solution (40% [w/v] PEG 4000 Sigma, 0.2 M Mannitol, 0.1 M CaCl2, DEPC treated water) is added and mixed gently prior to 20 min incubation in the dark. Then the samples are diluted with W5 solution and incubated for 15 min in the dark. The samples are centrifuged at 450xg with soft start/stop and resuspended in 1 ml WI solution (0.5 M Mannitol, 20 mM KCl, 4 mM MES pH = 5.7) before incubation in the dark. At each time-point, duplicate samples are independently transformed, for 14 samples per time-course, for hours 0, 6, 12, 24, 36, 48, 72 h post transformation. The samples for time 0 are frozen prior to addition of the WI, with the time of freezing set to the start of the time-course.

**Sample collection and DNA extraction.** Samples are pelleted using centrifugation at 450 x g, WI is removed and samples are flash frozen in liquid nitrogen. DNA is extracted using NucleoSpin Plant II extraction kit (Machery-Nagel Cat #) with modified protocol. Briefly, 600 ul PL1 lysis buffer are added to each sample with 10 ul RNAse and incubated 1 h at 65°C. After running through column, 675 ul PC buffer is added. Washing is done with 2 x with PW2 buffer before elution in 50 ul elution buffer.

**UMI-DSBseq adaptor design and preparation.** Design of the UMI-DSBseq adaptors was adapted from on the P7 tail of the xGen UDI-UMI adaptors from IDT, containing an 8 bp index for barcoding from the IDT8_i7 index list, and a 9 bp unique molecular identifier, for analysis with the recommended pipeline for single molecule sequencing. Two oligos are ordered from IDT as ULTRAMERs, a long tail containing P7 tail with UMI indexes, along with a final T at the 3' end with a phosphorothioate bond (*T) for sticky T/A ligation, as well as a semi-complementary oligo (P7 forward tail: GATCGGAA-GAGCGGGGACTATTTGC). The two oligos are annealed by heating to 95ºC and allowing to cool to room temperature. Prepared adaptors are diluted to 1 uM in 10 mM Tris pH = 8, and stored at -20ºC until use.

**UMI-DSB library preparation.** 20 ul of extracted DNA (25-50 ng) is restricted overnight with target-specific enzyme at 37ºC. After cleaning with magnetic beads, restricted DNA is end-repaired using DNA Polymerase 4. A-addition using Klenow is followed by UMI-DSBseq adaptor ligation using DNA Ligase 4 with 2 ul from each adaptor, in Quick Ligase buffer with cleaning between each step. Target-specific amplification is achieved using 1 primer specific to the target sequence (see table below) with a tail composed of the P5 illumina tail, and 1 primer identical to the 5' end of the P7 adaptor tail sequence from the UMI-DSBseq adaptors (CAAGCAGAAGACGGCATACGAGAT)[69]. The adaptor specific primer can bind only loci for which second strand synthesis was already achieved due to the elongation from the target-specific primer. In the final step, i5 indexes are added using an additional PCR with the short enrichment primers used in TruSeq Illumina library preparation. Following cleaning, concentration is measured using Qubit (Cat #) and evaluated for quality and size using TapeStation

**Next-generation sequencing of UMI-DSB libraries.** UMI-DSBseq libraries are sequenced using 150 bp paired-end Illumina Nextseq or Novaseq kits. Libraries were sequenced with either NextSeq or Novaseq Illumina machines at the Weizmann Institute using 300 bp paired-end kits. The run settings include 17 bp index 1 and 8 bp index 2, with 149–151 bp for each read 1 and read 2. Libraries are individual time-course samples are sequenced with 1–5 million reads per library.

**Control time-courses.** For each target, full control time-courses were developed either by transforming Cas9 only to WT protoplasts or by mock-transformation Cas9 expressing protoplasts. Processing proceeds as described above.

**In-vitro DSB assay.** Genomic DNA extracted from tomato protoplasts, equal to the quantity processed for samples in the experimental time-courses (40% of sample DNA from 200,000 protoplasts), is cleaved in-vitro with 40% of the RNP produced as described above−Following incubation at 37ºC for 1 h, EDTA is added to stop the reaction, and the cleaved gDNA is cleaned using magnetic beads, followed by processing through the UMI-DSBseq workflow, as described above.

**Nuclei isolation.** A nuclei isolation protocol was adapted from Doležel et al.[70]. To isolate protoplasts nuclei, 5 ml of freshly isolated protoplasts were centrifuged for 5 min at 450 g with a soft start. All the liquids were discarded, and the cell was resuspended in an ice-cold LB01 buffer and put on ice. After incubation of 10 min with occasional gentle shaking the liquid was filtered through a 42 um filter. DAPI was added to a final concentration of 2 ug/ml. samples were stored on ice till further processing. Isolating root nuclei−the ends (0.5 cm from the end of the root) of a few 14–16 day-old tomato seedlings grown as described previously were collected to a Petri dish containing 5 ml of LB01 buffer placed on ice. The roots were chopped, filtered through a 42 um filter and stained with DAPI, to a final concentration of 2 ug/ml. For analysis of transformation efficiency, GFP fused to Cas9 was obtained from IDT and transformed into protoplasts as described

above. Nuclei were extracted as described above with additional washing to remove remnants of RNP.

**Fluorescence assisted cell sorting (FACS).** The nuclei were analyzed using FACSAria III cell sorter (GB biosciences) with a 100 um nozzle at low speed. Gating was designed to count only single and round nuclei based on the measured forward scatter (FSC), side scatter (SSC), and DAPI fluorescence. The intensity of GFP and DAPI fluorescence was quantified from the selected population.

**Quantification and statistical analysis**

**Demultiplexing and generation of consensus sequences.** Demultiplexing is done using bcl2fassq, splitting the index 1 read into the UMI file and index file. Data processing pipeline was adapted from the IDT pipeline (https://www.youtube.com/watch?v=68sca_jsqg8&ab_channel=IntegratedDNATechnologies) for building consensus sequences using FGBIO.

Fastqs are aligned to a reference of the target sequence using bwa-mem (bwa mem -t 8 $ reference $ fastq_R1 $fastq_R2 > $ mapped_bam). Unmapped BAM files are generated using picard FastqToSam. The unmapped and mapped BAM files are merged using picard MergeBamAlignment (MAX_GAPS = -1, CLIP_ADAPTORS = true) and annotated with UMIs using FGBIO. Reads are grouped by UMI using FGBIOs GroupReadsByUmi (strategy = adjacency, edits = 1, min-map-q = 0,assign-tag = MI). Finally, consensus sequences are generated using FGBIO CallMolecularConsensusReads with min-reads = 2 and minimum input base quality set to 20. Final BAM files with consensus reads are converted to Fastqs using SamToFastq from picard and joined using ea-utils fastq-join.

**Quantification and characterization of consensus sequences.** The joined fastq files are processed through the UMI-DSBseq analysis pipeline (https://github.com/daniebt/UMI-DSBseq), and available as Jupyter notebook. The workflow requires as input the joined fastq files of the consensus sequences, and a sample sheet in.xlxs format containing the file name, sgRNA sequence, primer sequence, and amplicon sequence as well as relevant run information, for each sample in the time-course. The workflow proceeds as described in the following sections.

**Extracting WT reference sequence from sample sheet.** A 100 bp reference sequence (WT_REF) is extracted from the sample sheet, 50 bp upstream or downstream of the DSB site for each target. Fastq files are uploaded using the BioSeq SeqIO package, outputting a dictionary with a table of reads for each sample.

**Determining the State of the read as Intact, Indel containing, or DSB.** 12 bp indicators on each end of the 100 bp reference window are aligned to each read, using pairwise2 local alignment from the BioSeq package. A molecule is defined as Intact, if both left indicator and right indicator receive an alignment score of at least 10/12 bp. For each Intact molecule, the sequence between the two indicators (inclusive) is extracted for further analysis (seq_window). A molecule is defined as a DSBs, if only the left indicator receives the minimum alignment score. In the case of defined DSBs, region is extraction from the left indicator (inclusive) to the end of the reads (seq_window). Any reads that do not either criteria for Intact or DSB, or that contain more than 4 'N's in their sequence are defined as NA. PCR contamination is filtered by removing any molecules ending in a known primer sequence as opposed to the restricted end.

**Characterizing footprints of repair error.** seq_window for each read is aligned to the 100 bp WT_REF using pairwise local alignment from the BioSeq package. Intact reads are defined as WT if no gap was opened in either the read or the reference. If a gap is opened in the WT_REF, the

type of the molecule is defined as Insertion, and the state is changed to NHEJ. If a gap is open in the seq_window of the read, the Type of the molecule is defined as deletion and the state is changed to NHEJ. The name of the indel is defined by the missing or added bases prefixed by '+' or '−'. Deletions associated with microhomology at the site, are defined as MH_Del, and the identity of the microhomology is reported, defined as 2 or more bp of microhomology flanking the deletion.

**Characterizing DSBs by type and position.** the seq_window is aligned as with the Intact molecules, and 'N's at the end of the sequence are trimmed. Based on the size of the DSB, they are defined as precise, at the position 50 in the WT_REF, or either guide-side or PAM-side, if they are positioned to either side. In addition, if the DSB is longer than expected and contains non-matching sequences to the reference, it is categorized as 'Extended'. Characterized reads are grouped and counted.

**Kinetic model.** A set of ordinary differential equations was used to develop a model for calculating the rate constants between the different states in the process. The model is fit to the data using gradient descent optimization of the log-likelihood using the R package optmix. To constrain the optimization procedure, we imposed a maximum cutting rate of 10 cuts per hour, which largely exceeds the highest estimates for CRISPR-Cas9. To ensure that we converge to the global maximum, we initialize the optimization from 5.000 points sampled randomly following an exponential distribution with average 0.01 and within the constrained range of potential values. The log-likelihood is calculated assuming independence between molecules within each sample, so that the proportion of sampled molecules of the different types is expected to be proportional to the total proportion of molecules present and expected from the dynamical model. This results in a multinomial sampling, for which the likelihood can be computed exactly.

To control for errors resulting from sequencing or other steps of the UMI-DSBseq assay we used the control time courses (in which only intact molecules are expected) to estimate the probabilities that an intact molecule is classified as belonging to a different state. We then built error matrices specifying the probability that a molecule of any state is classified as that of another state under the assumption that intact molecules carrying indels would show the same error rates as intact molecules carrying the original sequence, and that the chance of a broken molecule to be read as intact is negligible. Note that the proportion of broken molecules is usually much lower than that of intact ones, so that the latter assumption is expected to have only minor effects. We explored other assumptions observing negligible effects. Such different implementations of the error matrix can be explored using the -e flag in the provided code. Additionally, for the 3-state model imprecise DSBs are modeled as errors arising from precise DSBs. Since precise DSBs are not found in the control time courses, the probability that such errors occur is estimated dynamically as an additional parameter in the optimization of the likelihood of each time course. The code for the model estimation can be found at https://github.com/fabrimafe/DSBtimecourse.

**Determining the parameters of the induction curve.** The accessibility of Cas9 and proportion of active RNPs over time, denoted as *RNP(t)*, are modeled by assuming that the rate of cutting depends on three parameters $U$, $r$ and $d$, co-estimated with the rate constants describing DNA repair. These parameters determine the shape of an induction curve, constituted by a logistic growth and the constant decay of the RNPs' activity, and constrained to have a $10^{-6}$ RNP activity at time 0. Specifically, $U$ represents the proportion of DNA molecules which remain uncut−for instance including in cells that are not transfected; $r$ the speed of induction; $d$ represents the decay of the RNPs' activity, whose half-life is $1/d$. Thus, under the assumption that only intact molecules can be cut, the activity of *RNP(t)* follows the function:

$$RNP(t) = (1 - U)*2^{-d*t}/(1 + \exp(-r*(t - \log 10^6/(r + 10^{-12})))) - (1 - [intact]), \quad (1)$$

set to 0 for RNP[t] < 0.

In both the 3-state and 4-state model the parameters $U$, $r$ and $d$ are estimated through numerical optimization of the likelihood. For the FACS experiment, we estimated $U$, $r$ and $d$ using a binomial fit with proportions as in Fig. S6G, assuming 30 min between transformation and the time of cell sorting. The estimate of the other parameters was then constrained using these values.

**Calculating confidence intervals.** Confidence intervals are calculated using a stratified bootstrap approach: 100 simulated datasets were generated by resampling the original data-points with replacement within each time point; for each of these datasets the optimization procedure was repeated and the 1% and 99% percentiles of each parameter estimates used as confidence intervals.

**Model comparison.** The Akaike information criterion was used for model comparisons. The relative likelihood was calculated from the AIC. To assess the stability of the model comparison, AIC and relative likelihood were computed for all bootstrap iterations. A proportion >95% of the bootstraps supporting the more complex model, i.e. with relative likelihood lower than 1 and lower AIC value for the more complex model, was used as a significance threshold.

**The 3-state model.** We implemented and compared two different dynamical models with different degree of complexity and detail, differing in the rates being modeled and in how DSBs ending at positions corresponding to the expected Cas9 cutting site and other unexpected DSBs, defined as precise and imprecise DSBs, respectively, are treated.

The *3-state model* follows the dynamics of intact molecules, DSBs and indels, without considering any biological difference between DSBs ending at positions corresponding to the expected Cas9 cutting site and other DSBs. Specifically, unexpected DSBs are modeled as errors introduced by the experimental processing or sequencing, which result in some DSBs ending at observed positions different than that corresponding to the expected Cas9 cutting site. Hence, precise DSBs (referred to as direct DSB) are observed as imprecise DSB (referred to as processed DSBs) with a given probability $e$. Alternatively, the parameter $e$ can be seen as a constant proportion of DSBs that end at slightly different positions (processed DSBs) but have the same dynamics as other DSBs (direct DSBs).

The systems of ordinary differential equations describing these models are shown below. The 3-state dynamical model can be represented as:

$$d[intact]/dt = -K_{cut}*RNP(t)*[intact] + P^*[DSB]$$
$$d[indels]/dt = E[DSB] \quad (2)$$
$$d[DSB]/dt = -(P + E)^*[DSB] + K_{cut}*RNP(t)*[intact]$$

where $K_{cut}$ is a constant cutting rate, $P$ and $E$ are the repair rates of DSBs towards intact molecules and indels, respectively. At each time $t$, the proportion of processed DSB is simply estimated as a proportion $e$ of the DSB. Hence:

[processed DSB] = [DSB] e
[direct DSB] = [DSB] (1-e).

For the 3-state model without repair we used the same model by fixing P to 0.

Concluding, the 3-state model is used to estimate four parameters ($e$, $K_{cut}$, $P$, $E$), in addition to the three parameters of the induction curve.

**The 4-state model.** A *4-state model*, which distinguishes between direct and processed DSBs. This more complex model, allows imprecise and precise DSBs to have different dynamics. Imprecise DSBs can appear through the successive molecular processing of precise DSBs. Both direct and processed DSBs can undergo precise repair to intact molecules (P) or error-prone (E) to indels, with rates $P_{direct}$ and $E_{direct}$, and $P_{processed}$ and $E_{processed}$, respectively. Direct DSB are converted to processed DSB with rate $K_{processing}$.

The 4-state model is:

$$d\,[\text{intact}] = -K_{cut}*RNP(t)*[\text{intact}] + P_{direct}*[\text{DSB}]$$
$$d\,[\text{indels}] = E_{direct}*[\text{DSB}] + E_{processed}*[\text{processed DSB}]$$
$$d\,[\text{DSB}] = -(P_{direct} + E_{direct} + K_{processing})*[\text{DSB}] + K_{cut}*RNP(t)*[\text{intact}]$$
$$d[\text{processed DSB}] = -(P_{processed} + E_{processed})*[\text{processed DSB}] + K_{processing}[\text{DSB}]$$

$$(3)$$

Concluding, the 4-state model is used to estimate six parameters ($K_{cut}$, $K_{processing}$, $P_{direct}$, $E_{direct}$, $P_{processed}$, $E_{processed}$), in addition to the three parameters of the induction curve.

**Calculating flow over the time course.** For each rate we calculated the number of molecules out of the ones initially present in the pool which underwent a specific transformation, e.g., from intact to precise DSBs with rate $k_{cut}$, through numerical integration of the systems of ordinary differential equations. Note that the total amount of molecules going through each rate can exceed 100%, e.g., more intact molecules than present at the beginning of the experiment can be cut to precise DSBs over the rate of the time course because some intact molecules are replenished via perfect repair. Numerical integration was computed at a resolution of 500 intervals each hour of the time course. The code for the numerical integration is deposited as an R script at github.com/fabrimafe/DSBtimecourse/plot_bootstraps.R

### Reporting summary
Further information on research design is available in the Nature Portfolio Reporting Summary linked to this article.

## Data availability
Resources and reagents are listed in Supplementary Table 12. The oligonucleotides used in this study are shown in Supplementary Data 1. Excel workbook of consensus sequence counts for each dataset in the study are shown in the Source Data. Sequencing data for the time courses have been deposited on NCBI, project PRJNA1113841. Flow cytometry data have been deposited on Zenodo, https://doi.org/10.5281/zenodo.11255717. Further information and requests for data, resources and reagents should be directed to and will be fulfilled by the lead contact, avi.levy@weizmann.ac.il. Source data are provided with this paper.

## Code availability
BASH scripts for calling consensus sequences and Jupyter notebooks for characterizing DSB and repair types for UM I-DSBseq data can be found at https://github.com/daniebt/UMI-DSBseq. The code to run the kinetic model and generate the estimates and figure in the paper can be found at www.github.com/fabrimafe/DSBtimecourse and Zenodo https://doi.org/10.5281/zenodo.11218185.

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

## Acknowledgements

We wish to thank the Israel Science Foundation ISF2332/19, and the Israeli Ministry of Innovation CRISPR-IL consortium to AAL for financial

support and the Clore Scholars Fund for the PhD fellowship for DBT; Dr. Barry Cohen for his assistance in optimizing the protoplast protocol; the protein expression unit at the Weizmann Institute for the Cas9 protein purification. We would like to thank Maayan Guetta, Ilan Hadad and Ido Sela for their technical assistance.

## Author contributions

This work was supervised by AL and CMB. DBT and CMB conceptualized and designed the UMI-DSBseq molecular assay. UMI-DSBseq analysis pipeline was written by DBT and AC. DBT, AH and CMB calibrated and optimized the RNP transformation protocol. DBT conducted experiments, library preparation and UMI-DSBseq analysis. Sequencing runs were carried out and managed by CMB. All kinetic modeling design, optimization, and implementation along with all statistical analysis was carried out by FM. Manuscript was written by DBT, FM, and AL.

## Competing interests

The authors declare no competing interests.

## Additional information

 16