## [Peer Review File · Nature Communications]

Reviewers' Comments:

Reviewer #1:

Remarks to the Author:

In this manuscript, Ben-Tov et al. describe a molecular and computational toolkit for direct quantification of DSB intermediates alongside repair-products through multiplexed single-molecule sequencing. They report the dynamics of DSB induction, end-processing, and repair at three CRISPR/Cas9 targets in tomato protoplasts, measuring the abundance of DSB intermediates and repair products. The authors present evidences that precise re-ligation is a prominent feature (40-70% of all repair events) and conclude that that editing efficiency in plants is determined, in some cases as much by the fidelity of the endogenous repair process as by the ability to efficiently induce DSBs at the target site.

The authors describe a molecular and computational toolkit for quantification of DSB intermediates at three different loci in the tomato genome. The method presented here has the potential to be of significance to the fields of DNA repair and CRISPR-Cas editing in plants. The models proposed resume well the observed data, however, their level of predictivity and usefulness for CRISPR-Cas strategies, is difficult to evaluate. The authors give evidences that correct re-ligation is a major feature of induced DSB repair, this result is in line with what has been observed in some animal models and is very significant to the field of plant DNA repair.

The general quality of the data is good and the data are well presented. The UMI-DSBseq methodology is technically well described and reaches expected standards. However, the proposed conclusions and models are not always well supported and would need additional evidences.

Different complementary experiments could support the conclusions proposed here:

- as mentioned by the authors, testing repair kinetics in NHEJ or HR mutants (or RNAi strategy), or using DNA repair inhibitors, would enable to confirm the models of repair and dynamic proposed here.
- Study of a mutant affected in cNHEJ would definitely confirm the important conclusion that correct re-ligation is a major feature of induced DSB repair.
- Of special importance would be the analysis of mutants affected in MMEJ repair as this repair pathway has been implicated in repair based on existing microhomology (as specified by the authors) but may also be involved in a subset of repair that does not involve pre-existing microhomology (for a review see Seol et al., 2017). Comparison of the output of repair between a cNHEJ mutant versus a MMEJ mutant would help decipher more precisely what is behind the "error-prone NHEJ". Status of the Psy1 locus in such mutants would be of particular interest as it seems that DSBs are correctly induced at this locus but with low indels production. Is precise repair particularly efficient at this locus or is MMEJ particularly inefficient?
- Use of an additional target showing high level of microhomology (4-5 bp) in the vicinity of the CRISPR-Cas9 DSB would confirm the role of MMEJ in the process of "error-prone NHEJ".
- Cas9 creates blunt ended DNA breaks near the PAM site whereas Cas12a generates staggered DNA breaks distal to the PAM site. It would be interesting to add an experiment using a Cas12a target, as this could potentially confirm and complement the models proposed in this study. Is correct re-ligation still a major feature of Cas12a induced DSB repair?

Minor comments:

- Figure 3A to 3F could the authors explain why positions of the peaks are slightly different between in vivo vs in vitro experiments?
- The authors followed the patterns of unrepaired DSBs versus NHEJ-mediated indel after transfection with RNPs at 3 different loci for 72 hours. What is the rationale for the 72h time-course? Why didn't the authors stop the time-course when the number of indels returned to the base line.

- What is the kinetics of the RNP (Cas9+sgRNA) in the protoplasts? This is an important information in order to discuss the kinetics of DSB during the time course. Can this kinetic vary from one RNP (Cas9 + one of the three guides) to another? A western blot analysis could help for this.
- What is the ratio of transfected protoplasts in the total population of protoplast? A GFP control could help for this. If not all the protoplasts are transfected, is this taken into consideration when estimating the number of "uncut or precisely repaired molecules"?
- Could the authors comment on the possibility that some Cas9-sgRNA complex would be still present after DNA extraction. Is there a risk that this could artificially increase the number of unrepaired DSBs observed?
- Could the authors comment on the possibility that potential DSB repair events that involved large ends resections would not be detected due to elimination of the primer specific to the target sequence or of the restriction site flanking the target?
- In order to estimate the robustness of the UMI-DSBseq strategy, one control experiment would be a targeted deep sequencing analysis of genomic DNA (no digestion) of the three different locus and a comparison of the mutations characterized by this technique to the indels and SNPs found via the UMI-DSBseq strategy.

Reviewer #2:

Remarks to the Author:

The manuscript by Ben-Tov et al., "Uncovering the Dynamics of Precise Repair at CRISPR/Cas9-induced Double-Strand Breaks", describes application of a new toolkit to evaluate dynamics of DSB formation and repair outcomes in a single cell. These questions are not only intriguing but also important for better understanding processes undergoing during genome editing experiments. The work was done based on a specific technical approach, it was well planned and executed, and results are quite solid.

Here are some of my thoughts and considerations I would like the authors to address, which can strengthen the manuscript, especially the Discussion part.

Since lines are not numbered, I would allow myself to list my comments/suggestion in a free order.

1. In the abstract, the authors state that 40-70% of all DSB are repaired via precise re-ligation. It is possible that I overlooked... but I did not see this statement/conclusion neither in the results not the discussion sections. I was left with some confusion about the statement.
2. Nevertheless, 40-70% perfect repair sounds correct. This reviewer came to the same conclusion by the analysis of various size deletions – the way to look at the fixed events different from the wild type of sequence. Such deletion sites are repaired exactly the same way and can be divided into categories – perfect re-ligation or junctions with Indels just like single DSB.
3. Since most groups evaluate cleavage activity (indirectly) via mutation frequency at the given target site, I would like the authors to provide the same evaluation for the three sites used in the study as control for their conclusions.
4. Psy1 site clearly behaves differently, my understanding, the authors conclude that its repair happens seamlessly with higher frequency than at the other two sites. It seems that the alternative explanation might be associated with lower cleavage activity of the site, please clarify.
5. Based on the authors analysis, would it be possible to make a prediction of which site(s) has higher/lower frequency of DSB repair without INDELS?
6. The delivery and analysis are performed on protoplasts, although this material can be considered for evaluation, one has to remember that protoplasts are very different from different explants used in transformation. This means that the dynamics of the DSB repair maybe different in different tissues, in cells that are actively proliferating or not undergoing cell division, etc.
7. Since the authors used delivery of RNP complexes, what if their half-life in tomato protoplasts? Has it been tested? Would the authors expect that with RNP delivery each site was cut and repaired only once or had potentials to be re-ligated and recut second time?
8. The authors are using term 'local off-target cleavage' that might be associated with not a blunt-

end cutting. In my opinion, this term is not very accurate and also rather confusing due to a very different meaning of off-target cleavage broadly used in the literature. The fact that the authors use word 'local' doesn't make it less confusing. I would recommend using a different terminology for the observed cleavage type.

9. Discussion section is full of speculations and potential applications, there is nothing wrong with that, but I would like it to be more connected to the results and discuss specifics of the experimental design, pluses and minuses of the approach, limitations, some of them I tried to describe above.

Reviewer #3:

Remarks to the Author:

The manuscript by Ben-Tov, Mafessoni et al. measures the kinetics of Cas9 repair in tomato protoplasts. DNA sequence data of 3 target sites over time are used to model rates of breakage and repair. This manuscript is interesting and builds on a prior report on Cas9 kinetics by improving the quantification with higher resolution data. Their UMI-DSBseq is a clever method to detect intact DNA, DSBs and indel fractions using next generation sequencing. Quantifying experimental data and extracting testable models from it is something that is needed in the field of gene editing field to unravel the mechanism of cutting and repair.

However, I am concerned that the authors in this paper base all their modelling and conclusions on the experimental data presented in Figure 2. In my opinion the experimental validation is too sparse for the authors' conclusions (see comments below).

Overall, I enjoyed reading this manuscript and would be happy to see a reworked version.

1. I am concerned that the model is based on only two independent experiments. Bootstrapping creates a new time series out of resampling with replacement, but is still based on only two independent experiments. For each time point there are only 2 points that can vary, which will not surprisingly lead to an apparent high confidence of the model. At minimum, the authors should expand the number of replicates.

2. The authors model the proportion of active RNPs by a decay function, but it is not clear on what the chosen function is based. E.g. how is the half-life determined? Also, it is not explained how they model the transfection efficiency. Both phenomena affect the modelling. Quantitation of Cas9 (e.g. by western blotting at the different time points), and measuring the transfection efficiency (e.g. co-transfection with GFP and determining the fluorescence) would help to validate the correctness of the model.

3. According to the model the indel fraction will accumulate in time eventually till all intact DNA has been processed. However, the authors' model does not address cell division, which could impact their modelling. Data in the paper are generated from an asynchronous culture of cells over a 72-hour time course, which could include several cell divisions. There are two potential issues: 1) daughter cells inheritance of RNPs complex (e.g. at a frequency less than 100%); and 2) DNA break repair suggests that cells with unresolved breaks arrest cell cycle. Both phenomena will decrease the fraction of the culture that can introduce new Cas9 breaks. Cells that do not resolve the break should arrest and so start temporarily "losing" against cells without a break, but once the break is resolved they re-enter the dividing population. This seems like it would affect both modelling of measurements of cut ends as well as uncut alleles. I would like to see the authors to discuss the potential impact of cell division.

4. The indel fraction in Figure 2A-C and in Figure S1J-L, of all targets shows a dip at t=72h compared to the total indels at t=48h. This is contrary to the assumption in the model that assumes that the indels are the endpoint of the model and will only accumulate in time. Moreover, the model fit as shown in Figure 4, S4, S5 also shows less correspondence to the last timepoints. Could this be due to a cell cycle arrest in the repairing cells? The authors should discuss this.

5. The authors state that they found that cleavage efficiency was high for all targets. The "high" cleavage efficiency statement is confusing because the half-lives of cutting at all three target sites exceeds multiple hours. What is considered "high" and what is considered "low" is a matter of

semantics, but it could be matched with the semantics of the current literature. E.g. Cutting efficiency by CRISPR have been associated with long cutting times and long residence times {Kim et al., 2014; Richardson et al., 2016; Brinkman et al., 2018}. The authors could explain their vision and model in Figure 7 more clearly.

Minor concerns:

6. The mathematical modelling methods section could be improved. I would like to see a precise definition for the parameters (U, r, decay) in RNPs decay model.

7. The authors argue that CRISPR editing efficacy is to lesser extent affected by cleavage efficiency. This statement is insufficiently experimentally validated, because only 3 targets for the DSBs are shown in the manuscript. All these targets are positioned in genes (most active genes are located in open chromatin areas). It has been reported that the chromatin structure is a major determinant of the Cas9 cutting efficiency {Chakrabarti et al., 2019; Gisler et al., 2019; Schep et al., 2019}.

8. The assay used to measure DSBs relies on primer binding site in the vicinity of the break. For the subset of broken molecules that are repaired by sister-chromatid HR, this strand will be resected (>kilobases) and thus invisible by PCR. One could argue that double cutting on both sisters may frustrate HR, which is why this is not a major concern. Nevertheless, the authors should discuss the limitations of their assay.

9. The representation of Figure 3 is confusing. The authors indicate that CRTISO has two primary DSBs (-3, -4) however this is not clearly visible in the plot. A barplot would represent this better, like in Figure 3G,H. Figure 3G,H shows too much repetition since it is the same data as in Figure 3B,E and can be removed.

10. What is represented by the green dashed line in Figure S1D-F?

11. The colour panel is confusing in Figure S1J-L because the same colour represents different indels for the different guides. I would suggest visualizing the indels in barplot (only t=72h).

Answers to Reviewer's Comments:

Reviewer #1 (Remarks to the Author)

In this manuscript, Ben-Tov et al. describe a molecular and computational toolkit for direct quantification of DSB intermediates alongside repair-products through multiplexed single-molecule sequencing. They report the dynamics of DSB induction, end-processing, and repair at three CRISPR/Cas9 targets in tomato protoplasts, measuring the abundance of DSB intermediates and repair products. The authors present evidences that precise re-ligation is a prominent feature (40-70% of all repair events) and conclude that that editing efficiency in plants is determined, in some cases as much by the fidelity of the endogenous repair process as by the ability to efficiently induce DSBs at the target site.

The authors describe a molecular and computational toolkit for quantification of DSB intermediates at three different loci in the tomato genome. The method presented here has the potential to be of significance to the fields of DNA repair and CRISPR-Cas editing in plants. The models proposed resume well the observed data, however, their level of predictivity and usefulness for CRISPR-Cas strategies, is difficult to evaluate. The authors give evidences that correct re-ligation is a major feature of induced DSB repair, this result is in line with what has been observed in some animal models and is very significant to the field of plant DNA repair.

The general quality of the data is good and the data are well presented. The UMI-DSBseq methodology is technically well described and reaches expected standards.

However, the proposed conclusions and models are not always well supported and would need additional evidences.

Different complementary experiments could support the conclusions proposed here:

1. as mentioned by the authors, testing repair kinetics in NHEJ or HR mutants (or RNAi strategy), or using DNA repair inhibitors, would enable to confirm the models of repair and dynamic proposed here. **We agree with reviewer #1 that testing repair kinetics in mutants would have added additional insight into the repair enzymes involved in precise repair. However, earlier work done in our lab has shown that these mutants are highly sterile (see picture below) and therefore we decided to focus on the kinetics of DSB formation and repair in the wildtype which by itself was a large amount of work. The reviewer's suggestion might be easier to implement in Arabidopsis, where these mutants are fertile. This is indeed an exciting avenue for future works and while we have begun working in that direction, our preliminary experiments suggest it will take much time to calibrate the system with regards to the RNP system and the protoplasting protocol for each mutant due to their variable growth phenotypes.**

NHEJ mutants in tomato

2. Study of a mutant affected in cNHEJ would definitely confirm the important conclusion that correct re-ligation is a major feature of induced DSB repair.

Thanks for this comment. This is absolutely correct, however, as mentioned above, cNHEJ tomato mutants are sterile and stunted and thus cannot be used here. However, in view of your comment, in order to be more careful, we replaced the use of correct re-ligation by correct ‘repair’ and we will discuss the likelihood that we are dealing probably with correct re-ligation. Indeed, the majority of protoplasts were characterized as being in G1 (see new Figure S5), supporting that repair is likely by NHEJ rather than by HR with a nearby sister chromatid. HR with a homolog is possible but less likely as there is no somatic pairing between homologs and we showed in several works that HR-repair is relatively infrequent (References 27, 28, 57 in the revised manuscript). This point is now directly addressed in the discussion.

3. Of special importance would be the analysis of mutants affected in MMEJ repair as this repair pathway has been implicated in repair based on existing microhomology (as specified by the authors) but may also be involved in a subset of repair that does not involve pre-existing microhomology (for a review see Seol et al., 2017). Comparison of the output of repair between a cNHEJ mutant versus a MMEJ mutant would help decipher more precisely what is behind the “error-prone NHEJ”. Status of the *Psy1* locus in such mutants would be of particular interest as it seems that DSBs are correctly induced at this locus but with low indels production. Is precise repair particularly efficient at this locus or is MMEJ particularly inefficient?

Again, we agree that the study of MMEJ using mutants with UMI-DSBseq would lead to insight into the types of pathways involved at these loci, and into MMEJ in general. However, as you can see in the above figure the *PolQ* mutant, which is involved in MMEJ, is highly sterile and stunted.

4. Use of an additional target showing high level of microhomology (4-5 bp) in the vicinity of the CRISPR-Cas9 DSB would confirm the role of MMEJ in the process of “error-prone NHEJ”.

The *Psy1* locus has high levels of microhomology including 5bp – CCTTG and 4bp – TGTT, flanking the DSB site (see Table of reference sequences in Supplementary File1). There is no footprint identified corresponding to the 5bp microhomology. Indels corresponding to the 4bp microhomology peak around 24 hours and account for 1-3% of total footprints. From this target, we can learn that the presence of microhomology near the break site is not necessarily associated with high levels of MMEJ-mediated repair error. We discuss this in the manuscript and expand in the discussion. See Supplementary File 1 and page 7 of the revised manuscript.

5. Cas9 creates blunt ended DNA breaks near the PAM site whereas Cas12a generates staggered DNA breaks distal to the PAM site. It would be interesting to add an experiment using a Cas12a target, as this could potentially confirm and complement the models proposed in this study. Is correct re-ligation still a major feature of Cas12a induced DSB repair?

We agree with reviewer #1 that exploring the dynamics of repair at Cas12a-induced staggered DSBs would add additional insight into precise repair. We attempted this in the revisions of the manuscript and obtained preliminary results. Unfortunately, we did not manage to calibrate well the experiment, and using Cas12 in our system resulted in a very minor proportion of cut molecules (little more than 1% of indels and DSBs, see top figure). This left us with very large uncertainty in the estimates (bottom plot), and unable to draw conclusions from this experiment. Thus we decided of not including this in the manuscript.

Minor comments:

1. Figure 3A to 3F could the authors explain why positions of the peaks are slightly different between in vivo vs in vitro experiments?

Thanks for picking this slight shift we hadn't noticed. We are not sure what exactly happened during the preparation of the line plot. However, since the data is more accurately described by barplots with discrete categories, we replaced Figure 3 with barplots and added Figure S3.

2. The authors followed the patterns of unrepaired DSBs versus NHEJ-mediated indel after transfection with RNPs at 3 different loci for 72 hours. What is the rationale for the 72h time-course? Why didn't the authors stop the time-course when the number of indels returned to the base line.

Thank you for this comment. The rationale for a 72-hour time-course was that during this time period, protoplasts did not divide, and we had evidence that there is still ongoing DSB formation activity. However, the FACS experiments indicate that some cells begin to die between 48-72 hours. As a result, and for the sake of consistency, we validated our conclusions by repeating the time-course experiments at high resolution for the first 24 hours. This maintains the analysis to the window of healthy cells and it confirms the evidence for precise repair, adding insight into earlier time points.

3. What is the kinetics of the RNP (Cas9+sgRNA) in the protoplasts? This is an important information in order to discuss the kinetics of DSB during the time course. Can this kinetic vary from one RNP (Cas9 + one of the three guides) to another? A western blot analysis could help for this.

Thanks. The RNP stability during the time course is an important point. We addressed it by analyzing fluorescent assisted cell sorting (FACS) of Cas9-GFP RNP transformed protoplasts (Figure S6), and showing that the vast majority of the cells contain large quantities of RNP, immediately upon transformation and through 48 hours. This confirms our finding of rapid DSB induction and very little RNP decay, likely due to the lack of cell divisions (Figure S5).

4. What is the ratio of transfected protoplasts in the total population of protoplast? A GFP control could help for this. If not all the protoplasts are transfected, is this taken into consideration when estimating the number of “uncut or precisely repaired molecules”?

This is indeed an important control that we had done while calibrating the system but not shown. We have repeated this control in a systematic manner, confirming that there was a very high transfection efficiency through the FACS analysis of GFP in protoplasts transformed with Cas9-GFP fusion throughout the experiment time course (Figure S6).

5. Could the authors comment on the possibility that some Cas9-sgRNA complex would be still present after DNA extraction. Is there a risk that this could artificially increase the number of unrepaired DSBs observed?

In order to prevent this, samples are flash frozen in liquid nitrogen and stored at -80C. Prior to thawing, the first step of DNA extraction is addition of lysis buffer containing high quantities of RNase prior incubation for 1 hour at 65 degrees Celsius. While some Cas9-sgRNA is likely present during the collection, it is likely that during the extraction, it cannot effectively cleave due to temperature and buffer.

6. Could the authors comment on the possibility that potential DSB repair events that involved large ends resections would not be detected due to elimination of the primer specific to the target sequence or of the restriction site flanking the target?

Large resections of both DNA strands might indeed be missed. However, resection that leaves 3' ends tails would not be missed: in order to ensure capturing of DSBs including those that have 3' overhangs, the UMI-DSBseq protocol includes an end-repair step, where the ends are filled in using T4 DNA polymerase.

7. In order to estimate the robustness of the UMI-DSBseq strategy, one control experiment would be a targeted deep sequencing analysis of genomic DNA (no digestion) of the three different locus and a comparison of the mutations characterized by this technique to the indels and SNPs found via the UMI-DSBseq strategy.

Thanks. UMI-DSBseq was compared to amplicon sequencing to confirm that the outcome of the indel analysis was comparable between the methods (Figure S1).

Reviewer #2 (Remarks to the Author)

The manuscript by Ben-Tov et al., “Uncovering the Dynamics of Precise Repair at CRISPR/Cas9-induced Double-Strand Breaks”, describes application of a new toolkit to evaluate dynamics of DSB formation and repair outcomes in a single cell. These questions are not only intriguing but also important for better understanding processes undergoing during genome editing experiments. The work was done based on a specific technical approach, it was well planned and executed, and results are quite solid.

Here are some of my thoughts and considerations I would like the authors to address, which can strengthen the manuscript, especially the Discussion part.

Since lines are not numbered, I would allow myself to list my comments/suggestion in a free order.

1. In the abstract, the authors state that 40-70% of all DSB are repaired via precise re-ligation. It is possible that I overlooked... but I did not see this statement/conclusion neither in the results not the discussion sections. I was left with some confusion about the statement.

Thanks for this request and sorry it was not well explained. The 40-70% refers to the percent of precise repair out of the total repair (the sum of precise and error-prone repair). In other words, from all the repair events, 40-70% were precise. This has been clarified in the Tables legend (see Repair accuracy in Table 1) and in the text.

2. Nevertheless, 40-70% perfect repair sounds correct. This reviewer came to the same conclusion by the analysis of various size deletions – the way to look at the fixed events different from the wild type of sequence. Such deletion sites are repaired exactly the same way and can be divided into categories – perfect re-ligation or junctions with Indels just like single DSB.

We thank reviewer #1 for emphasizing this point and have adjusted our discussion to emphasize this.

3. Since most groups evaluate cleavage activity (indirectly) via mutation frequency at the given target site, I would like the authors to provide the same evaluation for the three sites used in the study as control for their conclusions.

One of our main achievements was to distinguish between the cleavage rate and the final mutation frequency, namely repair error (or editing efficiency). We made this clearer by elaborating in the discussion when referring to Figure 8B.

4. *Psy1* site clearly behaves differently, my understanding, the authors conclude that its repair happens seamlessly with higher frequency than at the other two sites. It seems that the alternative explanation might be associated with lower cleavage activity of the site, please clarify.

At *Psy1*, as we wrote in point 3 above, the cleavage rate is of 64%, i.e. in the same range as with other loci. Low editing efficiency seems thus to be more related to repair fidelity than cleavage.

5. Based on the authors analysis, would it be possible to make a prediction of which site(s) has higher/lower frequency of DSB repair without INDELS?

No, we would need hundreds of targets to find such predictive features. This would be an exciting direction for future studies.

6. The delivery and analysis are performed on protoplasts, although this material can be considered for evaluation, one has to remember that protoplasts are very different from different explants used in transformation. This means that the dynamics of the DSB repair may be different in different tissues, in cells that are actively proliferating or not undergoing cell division, etc.

Thanks for this comment. We made sure to clarify that our conclusions are limited to our protoplast experimental system (end of discussion) and that, indeed, since the DNA repair machinery is specific to cell type and cell-cycle phase, it is likely that kinetics would differ between different tissues and conditions. Yet, we also mention that similar error-repair footprint types were obtained in protoplasts compared to transgenic plants when using the same guide.

7. Since the authors used delivery of RNP complexes, what if their half-life in tomato protoplasts? Has it been tested? Would the authors expect that with RNP delivery each site was cut and repaired only once or had potentials to be re-ligated and recut second time?

Thanks, we addressed this important issue, as mentioned in the answer to Referee#1, by analyzing fluorescent assisted cell sorting (FACS) of Cas9-GFP RNP transformed protoplasts (Figure S6), and showing that the vast majority of the cells contain large quantities of RNP, immediately upon transformation and through 48 hours. This confirms our finding of rapid DSB induction and very little RNP decay, likely due to the lack of cell divisions (Figure S5). Our data suggests that within the 72-hour window, there is sufficient time for 1-2 cycles of cutting and repair (see Table 1, Table S3).

8. The authors are using term ‘local off-target cleavage’ that might be associated with not a blunt-end cutting. In my opinion, this term is not very accurate and also rather confusing due to a very different meaning of off-target cleavage broadly used in the literature. The fact that the authors use word ‘local’ doesn’t make it less confusing. I would recommend using a different terminology for the observed cleavage type.

Thanks, we agree that the term off-target might be confusing. We changed the terminology to “a dual cleavage” pattern.

9. Discussion section is full of speculations and potential applications, there is nothing wrong with that, but I would like it to be more connected to the results and discuss specifics of the experimental design, pluses and minuses of the approach, limitations, some of them I tried to describe above.

We thank Reviewer #2 for the thoughtful comments and have modified the discussion based on this comment.

Reviewer #3 (Remarks to the Author):

The manuscript by Ben-Tov, Mafessoni et al. measures the kinetics of Cas9 repair in tomato protoplasts. DNA sequence data of 3 target sites over time are used to model rates of breakage and repair. This manuscript is interesting and builds on a prior report on Cas9 kinetics by improving the quantification with higher resolution data. Their UMI-DSBseq is a clever method to detect intact DNA, DSBs and indel fractions using next generation sequencing. Quantifying experimental data and extracting testable models from it is something that is needed in the field of gene editing field to unravel the mechanism of cutting and repair.

However, I am concerned that the authors in this paper base all their modelling and conclusions on the experimental data presented in Figure 2. In my opinion the experimental validation is too sparse for the authors' conclusions (see comments below).

Overall, I enjoyed reading this manuscript and would be happy to see a reworked version.

1. I am concerned that the model is based on only two independent experiments. Bootstrapping creates a new time series out of resampling with replacement, but is still based on only two independent experiments. For each time point there are only 2 points that can vary, which will not surprisingly lead to an apparent high confidence of the model. At minimum, the authors should expand the number of replicates.

Thanks for this comment. As suggested, we have repeated the experiment with four replicates and a high-resolution sampling. The results confirmed the previous finding of high DSB induction and precise repair. These results are now added in the section : “Validation of the precise repair of DSBs in high-resolution time courses”.

2. The authors model the proportion of active RNPs by a decay function, but it is not clear on what the chosen function is based. E.g. how is the half-life determined? Also, it is not explained how they model the transfection efficiency. Both phenomena affect the modelling. Quantitation of Cas9 (e.g by western blotting at the different time points), and measuring the transfection efficiency (e.g. co-transfection with GFP and determining the fluorescence) would help to validate the correctness of the model.

Our model infers this directly from the data. The details of this process were previously only described in the method section. To make this clearer we improved the method section and added a description of this step also in the main text. Additionally, we verified this using FACS of nuclei extracted from cells transformed with CRTISO Cas9-GFP RNPs (Figure S6). As mentioned to Reviewers 1 and 2, the FACS experiment suggests that the nuclei already contain RNPs at the point of time 0 collection. This supports rapid transport of the RNPs immediately, presumably during the incubation with the PEG solution (see methods). The results suggest that immediately following transformation, the vast majority of cells contain detectable levels of Cas9-GFP RNPs (Figure S6). This is maintained for 48 hours and validate the inferences made by the model.

3. According to the model the indel fraction will accumulate in time eventually till all intact DNA has been processed. However, the authors' model does not address cell division, which could impact their

modelling. Data in the paper are generated from an asynchronous culture of cells over a 72-hour time course, which could include several cell divisions. There are two potential issues: 1) daughter cells inheritance of RNPs complex (e.g. at a frequency less than 100%); and 2) DNA break repair suggests that cells with unresolved breaks arrest cell cycle. Both phenomena will decrease the fraction of the culture that can introduce new Cas9 breaks. Cells that do not resolve the break should arrest and so start temporarily “losing” against cells without a break, but once the break is resolved they re-enter the dividing population. This seems like it would affect both modelling of measurements of cut ends as well as uncut alleles. I would like to see the authors to discuss the potential impact of cell division.

Thanks for raising this potentially critical point which we have tested: We show in Figure S5 that protoplasts are almost exclusively in G1 and do not divide during the 72 hours of the experiment, unlike in actively dividing mammalian cells. This is regardless of DSB induction. Therefore, cell division is not a concern here. We discuss the potential effect of cell cycle and the aspects of the model system that are less ideal in the discussion.

4. The indel fraction in Figure 2A-C and in Figure S1J-L, of all targets shows a dip at $t=72\text{h}$ compared to the total indels at $t=48\text{h}$. This is contrary to the assumption in the model that assumes that the indels are the endpoint of the model and will only accumulate in time. Moreover, the model fit as shown in Figure 4, S4, S5 also shows less correspondence to the last timepoints. Could this be due to a cell cycle arrest in the repairing cells? The authors should discuss this.

Thanks for this comment. Evidence from our cell cycle analysis (Figure S5) and the literature suggests that protoplasts are themselves arrested due to the process of generating them (e.g. lack of cell wall). Cell cycle arrest is not due to the induction and repair process and instead is a characteristic of the model we work with. Regarding the last time points, the FACS experiments indicate that some cells begin to die between 48-72 hours. As a result, and for sake of consistency, we validated our conclusions by repeating the time-course experiments at high resolution for the first 24 hours. This maintains the analysis to the window of healthy cells and it confirms the evidence for precise repair, adding insight into earlier time points.

5. The authors state that they found that cleavage efficiency was high for all targets. The “high” cleavage efficiency statement is confusing because the half-lives of cutting at all three target sites exceeds multiple hours. What is considered “high” and what is considered “low” is a matter of semantics, but it could be matched with the semantics of the current literature. E.g. Cutting efficiency by CRISPR have been associated with long cutting times and long residence times {Kim et al., 2014; Richardson et al., 2016; Brinkman et al., 2018}. The authors could explain their vision and model in Figure 7 more clearly.

Thanks. Instead of using relative terminology, we are now referring to the observed numbers. We also made clear in the discussion that the lack of significant differences in cleavage efficiency might be due to a limited diversity of targets, to chromatin structure and other factors such as residency time and that cleavage efficiency must be considered also in addition to precise repair as a factor affecting editing efficiency. We also simplified the model (now Figure 8B).

Minor concerns:

6. The mathematical modelling methods section could be improved. I would like to see a precise definition for the parameters (U , r , decay) in RNPs decay model.

We improved this in the method section and in the main text.

7. The authors argue that CRISPR editing efficacy is to lesser extent affected by cleavage efficiency. This statement is insufficiently experimentally validated, because only 3 targets for the DSBs are shown in the manuscript. All these targets are positioned in genes (most active genes are located in open chromatin

areas). It has been reported that the chromatin structure is a major determinant of the Cas9 cutting efficiency {Chakrabarti et al., 2019; Gislser et al., 2019; Schep et al., 2019}.

Thanks, we have emphasized in the discussion that only exons were tested at only 3 targets and that cleavage efficiency might vary in different chromatin context and we added relevant references.

8. The assay used to measure DSBs relies on primer binding site in the vicinity of the break. For the subset of broken molecules that are repaired by sister-chromatid HR, this strand will be resected (>kilobases) and thus invisible by PCR. One could argue that double cutting on both sisters may frustrate HR, which is why this is not a major concern. Nevertheless, the authors should discuss the limitations of their assay.

This is an important comment. However, our cells are almost exclusively in G1 so precise repair via HR with a sister chromatid template is unlikely. HR with the homolog might be possible but is quite rare as we showed in planta (References 27, 28, 57 in the revised manuscript). These features of the assay are discussed in the revised version.

9. The representation of Figure 3 is confusing. The authors indicate that CRTISO has two primary DSBs (-3, -4) however this is not clearly visible in the plot. A barplot would represent this better, like in Figure 3G,H. Figure 3G,H shows too much repetition since it is the same data as in Figure 3B,E and can be removed.

Thanks, because of this comment and a similar comment by referee#1 we have replaced the line plot by a barplot that makes this more clear.

10. What is represented by the green dashed line in Figure S1D-F?

This is clarified in the legend of Figure S2.

11. The colour panel is confusing in Figure S1J-L because the same colour represents different indels for the different guides. I would suggest visualizing the indels in barplot (only t=72h).

Figure S1J-L was removed and replaced with tables indicating the top 10 indels at 48 hours at each target. See Figure S2.

Reviewers' Comments:

Reviewer #1:

Remarks to the Author:

In this resubmission of their manuscript Ben-Tov et al. describe a molecular and computational toolkit for direct quantification of DSB intermediates alongside repair-products through multiplexed single-molecule sequencing.

This revised version of the manuscript has been improved, and the various points I raised in the initial version have been appropriately addressed. Limitations in the feasibility of conducting the proposed experiments have been adequately justified. Additionally, all minor comments have been taken into consideration.

Regarding points 1 and 3, I understand that preparing protoplasts from mutants affected in the different DNA repair pathways is challenging due to the strong phenotypes of these mutants in tomato. It is intriguing to note that Arabidopsis and tomato DNA repair mutants exhibit dissimilar phenotypes. Perhaps, in the future, alternative strategies, such as RNAi approaches or dominant negative versions of DNA repair proteins, could be explored to validate the repair and dynamic models proposed in the manuscript. As suggested by the authors, testing the molecular and computational toolkit on other plant protoplasts could also yield valuable insights.

Regarding point 2 and the significance of cNHEJ as a major pathway for correct repair of the induced DSBs, I concur that HR with the homolog may only play a minor role in repair. However, the role of HR in protoplasts might differ from that in other somatic cells. To investigate this further, one approach could be to utilize a target site that exhibits SNPs between the two homologs (where the homolog containing the SNPs would not be cut). Detection of SNP copies after HR on the homolog should be feasible.

Regarding point 4, I agree that the Psy target serves as a good example for potential MMEJ repair. The observation that the presence of microhomology is not necessarily correlated with high levels of MMEJ-mediated repair is intriguing.

Regarding point 5, I appreciate that the authors tested the Cas12 enzyme. It is unfortunate that they encountered difficulties in calibrating the experiment, as a comparison with Cas9 would have provided valuable insights.

Minor comment:

- In the file 412760_1_data_set_8697726_s95hxx the table corresponding to CRTISO 24Hr is wrongly annotated (Psy in place of CRTISO).

Reviewer #2:

Remarks to the Author:

This is second revision of the previously revised manuscript by Ben-Tov et al., "Uncovering the Dynamics of Precise Repair at CRISPR/Cas9-induced Double-Strand Breaks", which describes application of a new toolkit to evaluate dynamics of DSB formation and repair outcomes in a single cell.

Overall, I am satisfied with changes/clarifications the authors provided to all three Reviewers' questions and recommendations. However, I have a few additional comments, which are not directly related to the experimental design and data analysis, but rather to some general statements the authors made, in my opinion, not quite accurately. I also have some general considerations for the Discussion, which might strengthen this part of the manuscript, and a few small editing suggestions.

Lines 14-15: "CRISPR/Cas9-mediated genome editing relies on error-prone repair of targeted DNA 15 double-strand breaks (DSBs). Understanding CRISPR/Cas9-mediated DSB induction...". I have

two comments: first, genome editing relies not only on error-prone repair but also on precise one when it comes to HDR, also on no DSB approaches – BE, PRIME editing (nickase), activation or suppression of gene expression and insertions using integrases and recombinases (dCas). I suggest going through the entire manuscript and correcting similar statements. Since the authors specifically studied the dynamics of cleavage and repair processes by NHEJ and INDELS introduction, this should be addressed. Second, I think that in two consecutive sentences it sounds 'clunky' to call CRISPR/Cas9-mediated genome editing and CRISPR/Cas9-mediated DSB induction.

Line 26: I am not certain that in the abstract you want to go to this level of details 'in two out of the three targets ...'. My immediate question – what is wrong/different with the third site?

Lines 28-29: '... suggesting an essential role of high-fidelity repair in limiting 29 CRISPR editing efficiency.' This goes back to my first comment and definition of genome editing. Therefore, limiting is too strong of a statement.

Line 43: I am not sure that term indel is fully accepted, as any abbreviation, it requires definition insertion and/or deletion (indel).

Line 90: I think this is the very first time when UMI-DSB-seq is introduced, also requires an explanation of what UNI stands for.

Lines 100-102: A 'clunky' sentence, which also has general term of genome editing... suggestion: Altogether, these results suggest that editing efficiency in plants is determined by the ability to efficiently induce DSBs at the target site and the fidelity of the endogenous repair process.

Line 123: the definition of UMI should go to the introduction.

Discussion section

General comments:

1. Evaluation of the repair mechanism and frequency of a DSB perfect recombination is a very important characteristic of a given TS. Quite often, cleavage efficiency evaluation is the most important parameter affecting overall frequency of both NHEJ-mediated and even more so of HDR-based genome editing. The authors somewhat mention this in lines 503-504 but I think it's important to discuss it in more detail to allow better understanding of the results and potential applications of the technology described in the manuscript.

2. I might have missed, and apologize if I have, a discussion related to RNP vs. DNA vector delivery. I think it is important to emphasize that the authors conclusions are the most relevant to i) cleavage efficiency evaluation and ii) NHEJ-based editing efficiency in case of RNP delivery. Constitutive expression of Cas9 nuclease and gRNA will lead to nearly 100% cleavage efficiency of a given TS. Also, when talking about indel introduction and consecutive knock out in plants, overall efficiency is very high and our ability to select for the intended outcome makes it very different/superior to many applications in mammalian and human systems.

Line 545: the same as above, lose use of genome editing term. Needs more specific definition.

Lines 556 and 591: G1 stage of cell cycle.

Lines 563-576: I agree with the authors' logic here as most labs are indeed using frequency of mutagenesis to measure TS cleavage efficiency. However, again, cleavage efficiency is even more important for naturally less efficient HDR-based genome editing applications. Therefore, the last sentence of the paragraph has the statement of '... our results show that precise repair plays an important role in shaping the editing efficiency of CRISPR/Cas9', is not quite accurate.

Lines 577 and 584: Again, I have problem with how the authors use term editing efficiency, I would like them to be more specific.

Reviewer #3:

Remarks to the Author:

The revised manuscript by Ben-Tov, Mafessoni et al. measures the kinetics of Cas9 repair at multiple loci in tomato protoplasts and uses that data to model rates of breakage and repair. This manuscript is improved from the previous submission, with new higher quality data (especially Figure 7), and textual changes that clarify the narrative.

The authors convince me of the value of the UMI-DSBseq method, which is a vast improvement of the method based on LM-PCR and also that theoretical models of cutting and repair can unravel more than just examine editing alone.

Reading this revision did not resolve my concern about the stability of the model and the conclusions that are related to that (see below). These are conceptual issues that can be addressed by discussion or modelling without collecting new data.

1) As I raised in the first review report, the confidence interval which is determined with the bootstrapping will give always a high value if there are 'only' 2 replicate experiments. The authors therefore performed additional replicates with high resolution sampling. Based on this new data, the absence of precise repair could not be rule out in 2 out of 3 loci. Therefore, I would like to see that the conclusions e.g. "the substantial proportion of precise repair (40-70%)" or "relatively high rates of precise repair", will be stated with more caution.

Also, I would not call this a "validation" experiment. Although, I value the newly produced data, it is a repetition of the same strategy with higher quality data. A validation experiment is performed with an independent method, so I would like to see the terminology changed.

2) The rate constants predicted by both models (24h, 72h) have some other discrepancies. For example, the newly predicted repair-error rates (E_{repair}) in Table 2, either doubled or was reduced by half and varied more than indicated by the confidence interval of Table 1. This raises a concern about the stability of the model. The window of the confidence interval in Table 1 may be too optimistic due to bootstrapping on only 2 time series. Though, the discrepancy in the rates may also be caused because the proportions of DSB and indels in the new short time series (24h, Figure 7) are higher in (Psy1, CRTISO) and lower (PhyB2) compared to those observed in Figure 4. Authors should address this variation. Could this be due to the transient transfection? How much is this variation affecting the values of the rate constants? I can image that the difference in concentration of the active RNP complex may change the K_{cut} .

3) Lastly, I wonder whether the large number of variables (in minimal 3-state model: 6 variables) that are predicted based on the time series data (intact, DSB, indel) detract from the accuracy of the key variables (e.g. rate constants for cutting and repair). The authors mention that besides the rates constants for cutting and repair, also the proportion of the active RNPs is inferred from the data. To verify this, they perform FACS experiment with Cas9-GFP. This experiment nicely shows the rapid transport of the RNPs and the almost complete transfection efficiency. However, in the model predictions of PhyB2, the U (untransfected fraction) is 0.4 h^{-1} , compared to 0 of the other loci. It is not expected that the transfection efficiency will vary between guides. The 0 value of U matches more the independent FACS experiment. Why don't the authors use the FACS experiment to infer/assume the U and the d . In my opinion the model would gain strength by using these parameters from completely different measurements (like the FACS experiment), which are not subject to the same 'noise' instead to rely on the prediction of these parameters.

Minor remarks.

- Missing reference in introduction after the sentence "50-55% precise repair in HeLa cells"
- Please state per locus (Psy1, CRTISO, PhyB2) how many replicates were performed for each time

series in Figures 2,4,7. E.g. in Figure 7, I can see that CRTISO has 4 replicates, but the other loci look to have 3. This is important to understand the value of the confidence interval.

- In 'dual cleavage at CRTISO' section there is wrong reference. "DSBs were induced in genomic DNA in vitro and compared to those characterized in the experimental time-course". Data is in Supplementary 3 instead of S2.

- Circular reasoning sentence in section "kinetic model reveal efficient DSB induction coupled with precise repair". "This is consistent with the mean proportion of indels directly observed in the 48 and 72-hour sampling points at the three targets." The consistency is of course logical, because the model was fitted on the same dataset.

- Supplementary Figure 2L: Indel -AAGGATT is misaligned in the figure. Breaksite annotation is not correct and in figure the indel is called -TAAGGAT

REVIEWER COMMENTS and Response

Reviewer #1 (Remarks to the Author):

In this resubmission of their manuscript Ben-Tov et al. describe a molecular and computational toolkit for direct quantification of DSB intermediates alongside repair-products through multiplexed single-molecule sequencing.

This revised version of the manuscript has been improved, and the various points I raised in the initial version have been appropriately addressed. Limitations in the feasibility of conducting the proposed experiments have been adequately justified. Additionally, all minor comments have been taken into consideration.

Regarding points 1 and 3, I understand that preparing protoplasts from mutants affected in the different DNA repair pathways is challenging due to the strong phenotypes of these mutants in tomato. It is intriguing to note that Arabidopsis and tomato DNA repair mutants exhibit dissimilar phenotypes. Perhaps, in the future, alternative strategies, such as RNAi approaches or dominant negative versions of DNA repair proteins, could be explored to validate the repair and dynamic models proposed in the manuscript. As suggested by the authors, testing the molecular and computational toolkit on other plant protoplasts could also yield valuable insights.

Thanks for approving the revisions and for the suggestions for future studies.

Regarding point 2 and the significance of cNHEJ as a major pathway for correct repair of the induced DSBs, I concur that HR with the homolog may only play a minor role in repair. However, the role of HR in protoplasts might differ from that in other somatic cells. To investigate this further, one approach could be to utilize a target site that exhibits SNPs between the two homologs (where the homolog containing the SNPs would not be cut). Detection of SNP copies after HR on the homolog should be feasible.

Thanks, this is definitely possible and we have plans (for which we already ran a pilot) for future work with the analysis of hybrids with SNPs at the DSB induction site.

Regarding point 4, I agree that the Psy target serves as a good example for potential MMEJ repair. The observation that the presence of microhomology is not necessarily correlated with high levels of MMEJ-mediated repair is intriguing.

Indeed, we were also surprised.

Regarding point 5, I appreciate that the authors tested the Cas12 enzyme. It is unfortunate that they encountered difficulties in calibrating the experiment, as a comparison with Cas9 would have provided valuable insights.

Minor comment:

- In the file 412760_1_data_set_8697726_s95hcx the table corresponding to CRTISO 24Hr is wrongly annotated (Psy in place of CRTISO).

Fixed

Reviewer #2 (Remarks to the Author):

This is second revision of the previously revised manuscript by Ben-Tov et al., “Uncovering the Dynamics of Precise Repair at CRISPR/Cas9-induced Double-Strand Breaks”, which describes application of a new toolkit to evaluate dynamics of DSB formation and repair outcomes in a single cell.

Overall, I am satisfied with changes/clarifications the authors provided to all three Reviewers' questions and recommendations. However, I have a few additional comments, which are not directly related to the

experimental design and data analysis, but rather to some general statements the authors made, in my opinion, not quite accurately. I also have some general considerations for the Discussion, which might strengthen this part of the manuscript, and a few small editing suggestions.

Lines 14-15: “CRISPR/Cas9-mediated genome editing relies on error-prone repair of targeted DNA 15 double-strand breaks (DSBs). Understanding CRISPR/Cas9-mediated DSB induction...”. I have two comments: first, genome editing relies not only on error-prone repair but also on precise one when it comes to HDR, also on no DSB approaches – BE, PRIME editing (nickase), activation or suppression of gene expression and insertions using integrases and recombinases (dCas). I suggest going through the entire manuscript and correcting similar statements. Since the authors specifically studied the dynamics of cleavage and repair processes by NHEJ and INDELS introduction, this should be addressed. Second, I think that in two consecutive sentences it sounds ‘clunky’ to call CRISPR/Cas9-mediated genome editing and CRISPR/Cas9-mediated DSB induction.

Thanks. We agree that the term Genome editing applies to many types of targeted DNA changes and we now made clear throughout the manuscript, that we focus on CRISPR/Cas9-mediated DSB induction and repair.

Line 26: I am not certain that in the abstract you want to go to this level of details ‘in two out of the three targets ...’. My immediate question – what is wrong/different with the third site?

Fixed

Lines 28-29: ‘... suggesting an essential role of high-fidelity repair in limiting 29 CRISPR editing efficiency.’ This goes back to my first comment and definition of genome editing. Therefore, limiting is too strong of a statement.

Fixed

Line 43: I am not sure that term indel is fully accepted, as any abbreviation, it requires definition insertion and/or deletion (indel).

Done

Line 90: I think this is the very first time when UMI-DSB-seq is introduced, also requires an explanation of what UNI stands for.

Done (Unique Molecular Identifier)

Lines 100-102: A ‘clunky’ sentence, which also has general term of genome editing... suggestion: Altogether, these results suggest that editing efficiency in plants is determined by the ability to efficiently induce DSBs at the target site and the fidelity of the endogenous repair process.

Fixed

Line 123: the definition of UMI should go to the introduction.

Done

Discussion section

General comments:

1. Evaluation of the repair mechanism and frequency of a DSB perfect recombination is a very important characteristic of a given TS. Quite often, cleavage efficiency evaluation is the most important parameter affecting overall frequency of both NHEJ-mediated and even more so of HDR-based genome editing. The authors somewhat mention this in lines 503-504 but I think it’s important to discuss it in more detail to allow better understanding of the results and potential applications of the technology described in the manuscript.

We have given a few examples of potential technological applications based on the new knowledge provided by UMI-DSB-seq following line 504.

2. I might have missed, and apologize if I have, a discussion related to RNP vs. DNA vector delivery. I think it

is important to emphasize that the authors conclusions are the most relevant to i) cleavage efficiency evaluation and ii) NHEJ-based editing efficiency in case of RNP delivery. Constitutive expression of Cas9 nuclease and gRNA will lead to nearly 100% cleavage efficiency of a given TS. Also, when talking about indel introduction and consecutive knock out in plants, overall efficiency is very high and our ability to select for the intended outcome makes it very different/superior to many applications in mammalian and human systems. We have described in the results (line 110-117) the rationale to use RNP in protoplasts, namely not relying on transcription and translation and offering a powerful and synchronized system for the study of repair kinetics. We now mention at the end of the discussions that the use of RNP rather than a transgenic system, limits the scope of our conclusions.

Line 545: the same as above, lose use of genome editing term. Needs more specific definition.

Fixed

Lines 556 and 591: G1 stage of cell cycle.

Fixed

Lines 563-576: I agree with the authors' logic here as most labs are indeed using frequency of mutagenesis to measure TS cleavage efficiency. However, again, cleavage efficiency is even more important for naturally less efficient HDR-based genome editing applications. Therefore, the last sentence of the paragraph has the statement of '... our results show that precise repair plays an important role in shaping the editing efficiency of CRISPR/Cas9', is not quite accurate.

Fixed

Lines 577 and 584: Again, I have problem with how the authors use term editing efficiency, I would like them to be more specific.

Fixed

Reviewer #3 (Remarks to the Author):

The revised manuscript by Ben-Tov, Mafessoni et al. measures the kinetics of Cas9 repair at multiple loci in tomato protoplasts and uses that data to model rates of breakage and repair.

This manuscript is improved from the previous submission, with new higher quality data (especially Figure 7), and textual changes that clarify the narrative.

The authors convince me of the value of the UMI-DSBseq method, which is a vast improvement of the method based on LM-PCR and also that theoretical models of cutting and repair can unravel more than just examine editing alone.

Reading this revision did not resolve my concern about the stability of the model and the conclusions that are related to that (see below). These are conceptual issues that can be addressed by discussion or modelling without collecting new data.

1) As I raised in the first review report, the confidence interval which is determined with the bootstrapping will give always a high value if there are 'only' 2 replicate experiments. The authors therefore performed additional replicates with high resolution sampling. Based on this new data, the absence of precise repair could not be rule out in 2 out of 3 loci. Therefore, I would like to see that the conclusions e.g. "the substantial proportion of precise repair (40-70%)" or "relatively high rates of precise repair", will be stated with more caution. Also, I would not call this a "validation" experiment. Although, I value the newly produced data, it is a repetition of the same strategy with higher quality data. A validation experiment is performed with an independent method, so I would like to see the terminology changed.

We toned this down as suggested and changed the terminology in respect to the validation.

2) The rate constants predicted by both models (24h, 72h) have some other discrepancies. For example, the

newly predicted repair-error rates (Erepair) in Table 2, either doubled or was reduced by half and varied more than indicated by the confidence interval of Table 1. This raises a concern about the stability of the model. The window of the confidence interval in Table 1 may be too optimistic due to bootstrapping on only 2 time series. Though, the discrepancy in the rates may also be caused because the proportions of DSB and indels in the new short time series (24h, Figure 7) are higher in (Psy1, CRTISO) and lower (PhyB2) compared to those observed in Figure 4. Authors should address this variation. Could this be due to the transient transfection? How much is this variation affecting the values of the rate constants? I can imagine that the difference in concentration of the active RNP complex may change the K_{cut} .

We thank the reviewer for pointing this out. The reviewer correctly points out that there is some between-batch variability that we do not discuss thoroughly in the manuscript. We believe this variability is likely due to both technical batch effects (efficiency and concentration of RNP complex, which exactly as the reviewer points out, will most likely affect K_{cut}) and intrinsic variability in the protoplast system, related to the status of the leaves when sampled, as well as in the protoplast extraction itself. In fact, we observed variation in the proportion of indels obtained (which is directly reflected in the different Erepair rates noticed by the reviewer) – also in other experiments – which points unequivocally to a real variability in induction (and possibly) repair. To exclude that this variation is due to our bootstrap procedure, or the model itself, we ran comprehensive sets of simulations, which we now report in part in the manuscript (Fig.S4, S15). Simulations show that our approach is very robust and accurate, despite the number of parameters, and has little false positives for precise repair. Thus, though the exact values of the different rates might subtly depend on experimental conditions, this does not impact our conclusions: we can certainly claim that there is substantial precise repair. However, we entirely agree that a certain degree of batch-effects exists, and it is hard to account for it entirely. Hence, we now acknowledge and emphasize this variability in the Discussion. We believe that future studies investigating this variability, as well variability in dynamics of precise repair in slightly different stress and growth conditions for plants/protoplast might help a lot in this respect. We would also like to add, in regard to the bootstrap procedure, that we also tested other methods for estimating uncertainty, including likelihood-ratio-based confidence intervals – which are calculated by default when using the pipeline. In different tests, the bootstrap was always the most conservative.

3) Lastly, I wonder whether the large number of variables (in minimal 3-state model: 6 variables) that are predicted based on the time series data (intact, DSB, indel) detract from the accuracy of the key variables (e.g. rate constants for cutting and repair). The authors mention that besides the rates constants for cutting and repair, also the proportion of the active RNPs is inferred from the data. To verify this, they perform FACS experiment with Cas9-GFP. This experiment nicely shows the rapid transport of the RNPs and the almost complete transfection efficiency. However, in the model predictions of PhyB2, the U (untransfected fraction) is 0.4 h^{-1} , compared to 0 of the other loci. It is not expected that the transfection efficiency will vary between guides. The 0 value of U matches more the independent FACS experiment. Why don't the authors use the FACS experiment to infer/assume the U and the d . In my opinion the model would gain strength by using these parameters from completely different measurements (like the FACS experiment), which are not subject to the same 'noise' instead to rely on the prediction of these parameters.

We thank a lot the reviewer for this suggestion. We already thought in fact of adding it to the paper, also because it complies with approaches that people used in the literature. So, we added this analysis in Figures S10,S14,S20,S21. Remarkably, all the estimated values are extremely similar whether the induction curve is estimated from the FACS (thus the 3-state model has only 3 variables) or whether we co-estimate it from the UMI-DSBseq data (6 variables). This shows that our results hold and are robust to this. We also highlight that it is correct that fixing the induction curve slightly reduces the uncertainty for some parameters; however, this reduction is only minor and we think that the slightly larger confidence intervals are worth to avoid the risk of misspecification of the induction curve parameters (for example, the induction curve estimated from the FACS might not be perfectly reliable because detected fluorescence might not completely correlate with RNP activity, or because between-experiment variability – which the reviewer correctly points out also in the previous comment).

We would also like to mention that the same is observed for the 4-state model, that has even more parameters; and that simulations show that for both 3 and 4-state model, we can infer repair parameters accurately even without imposing an induction curve.

Minor remarks.

- Missing reference in introduction after the sentence “50-55% precise repair in HeLa cells”

The reference was the same as for tobacco, at the end of the sentence. To make it clearer we added it for HeLa cells too.

- Please state per locus (Psy1, CRTISO, PhyB2) how many replicates were performed for each time series in Figures 2,4,7. E.g. in Figure 7, I can see that CRTISO has 4 replicates, but the other loci look to have 3. This is important to understand the value of the confidence interval.

Added. All time points for the 24-hour time courses (Fig.7) have 4 replicates, while all targets have 2 for the 72h time course. Some replicates have nearly the same values and thus they overlap, and for this reason they cannot be seen easily. We specified this in Figure 7 where this could create confusion.

- In ‘dual cleavage at CRTISO’ section there is wrong reference. “DSBs were induced in genomic DNA in vitro and compared to those characterized in the experimental time-course”. Data is in Supplementary 3 instead of S2.

We fixed the sentence, which was misleading (we do not have Supplementary File 3). We now only refer to Figure 3. Supplementary file 2 referred to the *in vivo* time course.

- Circular reasoning sentence in section “kinetic model reveal efficient DSB induction coupled with precise repair”. “This is consistent with the mean proportion of indels directly observed in the 48 and 72-hour sampling points at the three targets.” The consistency is of course logical, because the model was fitted on the same dataset.

This is correct, thanks. We reworded it as “..corresponding to..” to point this out and avoid the circularity.

- Supplementary Figure 2L: Indel –AAGGATT is misaligned in the figure. Breaksite annotation is not correct and in figure the indel is called -TAAGGAT

Thank you for noticing. We fixed the figure by aligning well the indels and renaming them accordingly.

Reviewer #3 (Remarks on code availability):

I glanced at the code, but I haven't run it. There was a README file with instructions. The code itself was not very well annotated.

Thanks for letting us know. We commented the code extensively. We also added code for simulations (which could be useful to test the code) and to fix the induction curve to a priori known curves, as used for the FACS.

We would also like to thank the reviewers for their thorough comments, which we believe truly helped us improving the manuscript.

Reviewers' Comments:

Reviewer #3:

Remarks to the Author:

My previous comments have been well addressed by the authors.

I spotted only a typo. On line 355, there is 0 missing "Erepair, is estimated between 0.34-0.86 of DSBs per hour". I believe this should be 0.034-0.086 h⁻¹

I have no further comments.

Response to Referees:

Reviewer #3 (Remarks to the Author):

My previous comments have been well addressed by the authors.

I spotted only a typo. On line 355, there is 0 missing "Erepair, is estimated between 0.34-0.86 of DSBs per hour". I believe this should be 0.034-0.086 h⁻¹

Thank you for picking this, we corrected the typo: It is now 0.034-0.086